# Algorithms with Prediction Portfolios

**Michael Dinitz**
Johns Hopkins University
mdinitz@cs.jhu.edu

**Sungjin Im**
UC Merced
sim3@ucmerced.edu

**Thomas Lavastida**[*]
University of Texas at Dallas
thomas.lavastida@utdallas.edu

**Benjamin Moseley**
Carnegie Mellon University
moseleyb@andrew.cmu.edu

**Sergei Vassilvitskii**
Google Research
sergeiv@google.com

## Abstract

The research area of algorithms with predictions has seen recent success showing how to incorporate machine learning into algorithm design to improve performance when the predictions are correct, while retaining worst-case guarantees when they are not. Most previous work has assumed that the algorithm has access to a single predictor. However, in practice, there are many machine learning methods available, often with incomparable generalization guarantees, making it hard to pick the best method a priori. In this work we consider scenarios where multiple predictors are available to the algorithm and the question is how to best utilize them.

Ideally, we would like the algorithm's performance to depend on the quality of the *best* predictor. However, utilizing more predictions comes with a cost, since we now have to identify which prediction is the best. We study the use of multiple predictors for a number of fundamental problems, including matching, load balancing, and non-clairvoyant scheduling, which have been well-studied in the single predictor setting. For each of these problems we introduce new algorithms that take advantage of multiple predictors, and prove bounds on the resulting performance.

## 1 Introduction

An exciting recent line of research attempts to go beyond traditional worst-case analysis of algorithms by equipping algorithms with *machine-learned predictions*. The hope is that these predictions allow the algorithm to circumvent worst case lower bounds when the predictions are good, and approximately match them otherwise. The precise definitions and guarantees vary with different settings, but there have been significant successes in applying this framework for many different algorithmic problems, ranging from general online problems to classical graph algorithms (see Section 1.2 for a more detailed discussion of related work, and [33] for a survey). In all of these settings it turns out to be possible to define a "prediction" where the "quality" of the algorithm (competitive ratio, running time, etc.) depends the "error" of the prediction. Moreover, in at least some of these settings, it has been further shown that this prediction is actually learnable with a small number of samples, usually via standard ERM methods [18].

Previous work has shown the power of accurate predictions, and there are numerous examples showing improved performance in both theory and practice. However, developing accurate predictors remains an art, and a single predictor may not capture all of the subtleties of the instance space. Recently, researchers have turned to working with *portfolios of predictors*: instead of training a single model, train multiple models, with the hope that one of them will give good guarantees.

---

[*]Work was done while the author was at Carnegie Mellon University.

36th Conference on Neural Information Processing Systems (NeurIPS 2022).

It is easy to see why the best predictor in a portfolio may be *significantly* better than a one-size fits all predictor. First, many of the modern machine learning methods come with a slew of hyperparameters that require tuning. Learning rate, mini-batch size, optimizer choice, all of these have significant impact on the quality of the final solution. Instead of commiting to a single setting, one can instead try to cover the parameter space, with the hope that some of the predictors will generalize better than others. Second, problem instances themselves may come from complex distributions, consisting of many latent groups or clusters. A single predictor is forced to perform well on average, whereas multiple predictors can be made to "specialize" to each cluster.

In order to take advantage of the increased accuracy provided by the portfolio approach, we must adapt algorithms with predictions to take advantage of multiple predictions. To capture the gains in performance, the algorithm must perform as if equipped with the best predictor, auto-tuning to use the best one available in the portfolio. However, it is easy to see that there should be a cost as the size of the portfolio grows. In the extreme, one can add every possible prediction to the portfolio, providing no additional information, yet now requiring high performance from the algorithm. Therefore, we must aim to minimize the dependence on the number of predictions in the portfolio.

We remark that the high level set up may be reminiscent of expert- or bandit-learning literature. However, there is a critical distinction. In expert and bandit learning, we are given a sequence of problem instances, and the goal is to compete (minimize regret) with respect to the best prediction *averaged* over the whole sequence. On the other hand, in our setup, we aim to compete with the best predictor on a *per-instance* basis.

**Previous work on multiple predictions.** Bhaskara et al. studied an online linear optimization problem where the learner seeks to minimize the regret, provided access to multiple hints [16]. Inspired by the work, Anand et al. recently studied algorithms with multiple learned predictions in [7], proving strong bounds for important online covering problems including online set cover, weighted caching, and online facility location. It was a significant extension of the work [22] which studied the rent-or-buy problem with access to two predictions. However, their techniques and results are limited to online covering problems. Moreover, they do not discuss the learning aspects at all: they simply assume that they are given $k$ predictions, and their goal is to have competitive ratios that are based on the minimum error of any of the $k$ predictions. (They actually compete against a stronger dynamic benchmark, but for our purposes this distinction is not important.)

On the other hand Balcan et al. [14] look at this problem through a data driven algorithm lens and study the sample complexity and generalization error of working with $k$ (as opposed to 1) parameter settings. The main difference from our work is that they also aim learn a selector, which selects one of the $k$ parameters *prior* to beginning to solve the problem instance. In contrast, in this work we make the selection during the course of the algorithm, and sometimes switch back and forth while honing in on the best predictor.

## 1.1 Our Results and Contributions

In this paper we study three fundamental problems, min-cost perfect matching, online load balancing, and non-clairvoyant scheduling for total completion time, in this new setting. Each of these has seen significant success in the single-prediction model but is not covered by previous multiple-prediction frameworks. Our results are primarily theoretical, however we have included a preliminary empirical validation of our algorithm for min-cost perfect matching in the supplementary material.

For each of these we develop algorithms whose performance depends on the error of the *best* prediction, and explore the effect of the number of predictions, $k$. Surprisingly, in the case of matching and scheduling we show that using a limited number of predictions is essentially free, and has *no* asymptotic impact on the algorithm's performance. For load balancing, on the other hand, we show that the cost of multiple predictions grows *logarithmically* with $k$, again implying a tangible benefit of using multiple predictions. We now describe these in more detail.

**Min-Cost Perfect Matching.** We begin by showcasing our approach with the classical min-cost perfect matching problem in Section 3. This problem was recently studied by [17, 18] to show that it is possible to use learned predictions to improve *running times* of classical optimization problems. In particular, [18] showed it is possible to speed up the classical Hungarian algorithm by predicting dual values, and moreover that it is possible to efficiently (PAC-)learn the best duals. We show that

simple modifications of their ideas lead to similar results for multiple predictions. Interestingly, we show that as long as $k \leq O(\sqrt{n})$, the extra "cost" (running time) of using $k$ predictions is negligible compared to the cost of using a single prediction, so we can use up to $\sqrt{n}$ predictions "for free" while still getting running time depending on the best of these predictions. Moreover, since in this setting running time is paramount, we go beyond sample complexity to show that it is also computationally efficient to learn the best $k$ predictions.

**Online Load Balancing with Restricted Assignments.** We continue in Section 4 with the fundamental load balancing problem. In this problem there are $m$ machines, and $n$ jobs which appear in online fashion. Each job has a size, and a subset of machines that it can be assigned to. The goal is to minimize the maximum machine load (i.e., the makespan). This problem has been studied extensively in the traditional scheduling and online algorithms literature, and recently it has also been the subject of significant study given a single prediction [26–28]. In particular, Lattanzi, Lavastida, Moseley, and Vassilvitskii [26] showed that there exist per machine "weights" and an allocation function so that the competitive ratio of the algorithm depends logarithmically on the maximum error of the predictions. We show that one can use $k$ predictions and incur an additional $O(\log k)$ factor in the competitive ratio, while being competitive with the error of the *best* prediction. Additionally, we show that learning the best $k$ predicted weights (in a PAC sense) can be done efficiently.

**Non Clairvoyant Scheduling** Finally, in Section 5 we move to the most technically complex part of this paper. We study the problem of scheduling $n$ jobs on a single machine, where all jobs are released at time 0, but where we do not learn the length of a job until it actually completes (the *non-clairvoyant* model). Our objective is to minimize the sum of completion times. This problem has been studied extensively, both with and without predictions [24, 30, 35, 37]. Most recently, Lindermayr and Megow [30] suggested that we use an *ordering* as the prediction (as opposed to the more obvious prediction of job sizes), and use the difference between the cost induced by the predicted ordering and the cost induced by the instance-optimal ordering as the notion of "error". In this case, simply following the predicted ordering yields an algorithm with error equal to the prediction error.

We extend this to the multiple prediction setting, which turns out to be surprisingly challenging. The algorithm of [30] is quite simple: follow the ordering given by the prediction (and run a 2-competitive algorithm in parallel to obtain a worst-case backstop). But we obviously cannot do this when we are given multiple orderings! So we must design an algorithm which considers all $k$ predictions to build a schedule that has error comparable to the error of the *best* one. Slightly more formally, we prove that we can bound the sum of completion times by $(1 + \epsilon)\text{OPT}$ plus $\text{poly}(1/\epsilon)$ times the error of the best prediction, under the mild assumption that no set of at most $\log \log n$ jobs has a large contribution to OPT.

To do this, we first use sampling techniques similar to those of [24] to estimate the size of the approximately $\epsilon n$'th smallest job without incurring much cost. We then use even more sampling and partial processing to determine for each prediction whether its $\epsilon n$ prefix has many jobs that should appear later (a bad sequence) or has very few jobs that should not be in the prefix (a good sequence). If all sequences are bad then every prediction has large error, so we can use a round robin schedule and charge the cost to the prediction error. Otherwise, we choose one of the good orderings and follow it for its $\epsilon n$ prefix (being careful to handle outliers). We then recurse on the remaining jobs.

## 1.2 Related Work

As discussed, the most directly related papers are Anand et al. [7] and Balcan, Sandholm, and Vitercik [14]; these give the two approaches (multiple predictions and portfolio-based algorithm selection) that are most similar to our setting. The single prediction version of min-cost bipartite matching was studied in [17, 18], the single prediction version of our load balancing problem was considered by [26–28] (and a different though related load balancing problem was considered by [4]), and the single prediction version of our scheduling problem was considered by [30] with the same prediction that we use (an ordering) and earlier with different predictions by [24, 37, 39]. Online scheduling with estimates of the true processing times was considered in [11, 12].

More generally, there has been an enormous amount of recent progress on algorithms with predictions. This is particularly true for online algorithms, where the basic setup was formalized by [31] in the

context of caching. For example, the problems considered include caching [25, 31, 38], secretary problems [9, 20], ski rental [5, 37, 39], and set cover [15]. There has also been recent work on going beyond traditional online algorithms, including work on running times [17, 18], algorithmic game theory [2, 21, 32], and streaming algorithms [1, 19, 23]. The learnability of predictions for online algorithms with predictions was considered by [6]. They give a novel loss function tailored to their specific online algorithm and prediction, and study the sample complexity of learning a mapping from problem features to a prediction. While they are only concerned with the sample complexity of the learning problem, we also consider the computational complexity, giving polynomial time $O(1)$-approximate algorithms for the learning problems associated with min-cost matching and online load balancing.

The above is only a small sample of the work on algorithms with predictions. We refer the interested reader to a recent survey [33], as well as a recently set up website which maintains a list of papers in the area [29].

## 2 Learnability

When designing new methods in the algorithms with predictions setting, the predictions under consideration must satisfy two constraints. First, they should be useful to the algorithm, so that using the predictions allows the algorithm to achieve better running time, competitive ratio, or some other performance measure. Second, they must be *learnable*: it must be feasible to find good predictions given a set of problem instances.

To rigorously prove learnability, we follow previous work [13, 18, 34] and focus on proving a bound on the sample complexity of finding the best predictions that generalize.

Our main result shows that for a given problem, the pseudo-dimension of finding $k$ predictions is $\tilde{O}(k)^2$ factor larger than that for finding a single best predictor. We state the formal Theorem below, but defer the proof to the supplementary material.

**Theorem 2.1.** *Let $\mathcal{F}$ be a class of functions $f : \mathcal{X} \to \mathbb{R}$ with pseudo-dimension $d$ and let $\mathcal{F}^k := \{F(x) = \min_{\ell \in [k]} f^\ell(x) \mid f^1, f^2, \ldots, f^k \in \mathcal{F}\}$. Then the pseudo-dimension of $\mathcal{F}^k$ is at most $\tilde{O}(dk)$.*

Note that this directly implies that the sample complexity when looking for $k$ predictions is a factor of $k$ larger than that of a single predictor by the following well-known theorem.

**Theorem 2.2.** *[8, 34, 36] Let $\mathcal{D}$ be a distribution over a domain $X$ and $\mathcal{F}$ be a class of functions $f : \mathcal{X} \to [0, H]$ with pseudo-dimension $d_{\mathcal{F}}$. Consider $S$ independent samples $x_1, x_2, \ldots, x_S$ from $\mathcal{D}$. There is a universal constant $c_0$, such that for any $\epsilon > 0$ and $\delta \in (0, 1)$, if $S \geq c_0 \left(\frac{H}{\epsilon}\right)^2 (d_{\mathcal{F}} + \ln(1/\delta))$ then we have*

$$\left| \frac{1}{s} \sum_{s=1}^{S} f(x_i) - \mathbb{E}_{x \sim \mathcal{D}}[f(x)] \right| \leq \epsilon$$

*for all $f \in \mathcal{F}$ with probability at least $1 - \delta$.*

## 3 Minimum Cost Bipartite Matching with Predicted Duals

In this section we study the minimum cost bipartite matching problem with multiple predictions. The case of a single prediction has been considered recently [17, 18], where they used dual values as a prediction and showed that the classical Hungarian algorithm could be sped up by using appropriately learned dual values. Our goal in this section is to extend these results to multiple predictions, i.e., multiple duals. In particular, in Section 3.2 we show that we can use $k$ duals and get running time comparable to the time we would have spent if we used the single best of them in the algorithm of [18], with no asymptotic loss if $k$ is at most $O(\sqrt{n})$. Then in Section 3.3 we show that $k$ predictions can be learned with not too many more samples (or running time) than learning a single prediction.

---

[2] $\tilde{O}(\cdot)$ suppresses logarithmic factors

## 3.1 Problem Definition and Predicted Dual Variables

In the minimum cost bipartite matching problem we are given a bipartite graph $G = (V, E)$ with $n = |V|$ vertices and $m = |E|$ edges, with edge costs $c \in \mathbb{Z}^E$. The objective is to output a perfect matching $M \subseteq E$ which minimizes the cost $c(M) := \sum_{e \in E} c_e$. This problem is exactly captured by the following primal and dual linear programming formulations.

Dinitz et al. [18] studied initializing the Hungarian algorithm with a prediction $\hat{y}$ of the optimal dual solution $y^*$. They propose an algorithm which operates in two steps. First, the predicted dual solution $\hat{y}$ may not be feasible, so they give an $O(n + m)$ time algorithm which recovers feasibility (which we refer to as Make-Feasible). Second, the now-feasible dual solution is used in a primal-dual algorithm such as the Hungarian algorithm (which we refer to as Primal-Dual) and they show that the running time depends on the $\ell_1$ error in the predicted solution. In addition to this they show that learning a good initial dual solution is computationally efficient with low sample complexity. More formally, they proved the following theorems.

$$\min \quad \sum_{e \in E} c_e x_e \quad \text{(MWPM-P)}$$
$$\sum_{e \in N(i)} x_e = 1 \quad \forall i \in V$$
$$x_e \geq 0 \quad \forall e \in E$$

$$\max \quad \sum_{i \in V} y_i \quad \text{(MWPM-D)}$$
$$y_i + y_j \leq c_e \quad \forall e = ij \in E$$

**Theorem 3.1** (Dinitz et al. [18]). *Let $(G, c)$ be an instance of minimum cost bipartite matching and $\hat{y}$ be a prediction of an optimal dual solution $y^*$. There exists an algorithm which returns an optimal solution and runs in time $O(m\sqrt{n} \cdot \|y^* - \hat{y}\|_1)$.*

**Theorem 3.2** (Dinitz et al. [18]). *Let $\mathcal{D}$ be an unknown distribution over instances $(G, c)$ on $n$ vertices and let $y^*(G, c)$ be an optimal dual solution for the given instance. Given $S$ independent samples from $\mathcal{D}$, there is a polynomial time algorithm that outputs a solution $\hat{y}$ such that*

$$\mathbb{E}_{(G,c) \sim \mathcal{D}} \left[ \|y^*(G, c) - \hat{y}\|_1 \right] \leq \min_y \mathbb{E}_{(G,c) \sim \mathcal{D}} \left[ \|y^*(G, c) - y\|_1 \right] + \epsilon$$

*with probability $1 - \delta$ where $S = \text{poly}(n, \frac{1}{\epsilon}, \frac{1}{\delta})$.*

## 3.2 Using $k$ Predicted Dual Solutions Efficiently

Given $k$ predicted dual solutions $\hat{y}^1, \hat{y}^2, \ldots, \hat{y}^k$, we would like to efficiently determine which solution has the minimum error for the given problem instance. Note that the predicted solutions may still be infeasible and that we do not know the target optimal dual solution $y^*$. We propose the following simple algorithm which takes as input $k$ predicted solutions and whose running time depends only on the $\ell_1$ error of the *best* predicted solution. First, we make each predicted solution feasible, just as before. Next, we select the (now-feasible) dual solution with highest dual objective value and proceed running the primal-dual algorithm with only that solution. See Algorithm 1 for pseudo-code.

We have the following result concerning Algorithm 1. To interpret this result, note that the cost for increasing the number of predictions is $O(k(n+m))$, which will be dominated by the $m\sqrt{n}$ term we pay for running the Hungarian algorithm unless $k$ is extremely large (certainly larger than $\sqrt{n}$) or there is a prediction with 0 error

---

**Algorithm 1** Minimum cost matching with $k$ predicted dual solutions

1: **procedure** $k$-PREDICTEDPRIMAL-DUAL$(G, c, \hat{y}^1, \hat{y}^2, \ldots, \hat{y}^k)$
2:     **for** $\ell \in [k]$ **do**
3:         $y^\ell \leftarrow$ MakeFeasible$(G, c, \hat{y}^\ell)$
4:     **end for**
5:     $\ell' \leftarrow \arg\max_{\ell \in [k]} \sum_{i \in V} y_i^\ell$
6:     $M \leftarrow$ Primal-Dual$(G, c, y^{\ell'})$
7:     Return $M$
8: **end procedure**

---

(which is highly unlikely). Hence we can reap the benefit of a large number of predictions "for free".

**Theorem 3.3.** *Let $(G, c)$ be a minimum cost bipartite matching instance and let $\hat{y}^1, \hat{y}^2, \ldots, \hat{y}^k$ be predicted dual solutions. Algorithm 1 returns an optimal solution and runs in time $O(k(n + m) + m\sqrt{n} \cdot \min_{\ell \in [k]} \|y^* - \hat{y}^\ell\|_1)$.*

We defer the proof to the supplementary material. But correctness is essentially direct from [18], and the running time requires just a simple modification of the analysis of [18].

### 3.3 Learning $k$ Predicted Dual Solutions

Next we extend Theorem 3.2 to the setting where we output $k$ predictions. Let $\mathcal{D}$ be a distribution over problem instances $(G, c)$ on $n$ vertices. We show that we can find the best set of $k$ predictions. More formally, we prove the following theorem.

**Theorem 3.4.** *Let $\mathcal{D}$ be an unknown distribution over instances $(G, c)$ on $n$ vertices and let $y^*(G, c)$ be an optimal dual solution for the given instance. Given $S$ independent samples from $\mathcal{D}$, there is a polynomial time algorithm that outputs $k$ solutions $\hat{y}^1, \hat{y}^2, \ldots, \hat{y}^k$ such that*

$$\mathbb{E}_{(G,c)\sim\mathcal{D}}\left[\min_{\ell\in[k]}\|y^*(G, c) - \hat{y}^\ell\|_1\right] \leq O(1) \cdot \min_{y^1,y^2,\ldots,y^k}\mathbb{E}_{(G,c)\sim\mathcal{D}}\left[\min_{\ell\in[k]}\|y^*(G, c) - y^\ell\|_1\right] + \epsilon$$

*with probability $1 - \delta$ where $S = \mathrm{poly}(n, k, \frac{1}{\epsilon}, \frac{1}{\delta})$.*

The proof of this theorem can be found in the supplementary material, but it is straightforward. The sample complexity is due to combining Theorem 2.1 with Theorem 3.2 (or more precisely, with the pseudo-dimension bound which implies Theorem 3.2). The $O(1)$-approximation factor and polynomial running time is from the observation that the ERM problem in this setting is just an instance of the $k$-median clustering problem.

## 4 Online Load Balancing with Predicted Machine Weights

We now apply our framework to online load balancing with restricted assignments. In particular, we consider proportional weights, which have been considered in prior work [26–28]. Informally, we show in Section 4.2 that if $\beta$ is the cost of the *best* of the $k$ predictions, then even without knowing a priori which prediction is best, we get cost of $O(\beta \log k)$. Then in Section 4.3 we show that it does not take many samples to actually learn the best $k$ predictions.

### 4.1 Problem Definition and Proportional Weights

In online load balancing with restricted assignments there is a sequence of $n$ jobs which must be assigned to $m$ machines in an online fashion. Upon seeing job $j$, the online algorithm observes its size $p_j > 0$ and a neighborhood $N(j) \subseteq [m]$ of *feasible* machines. The algorithm must then choose some feasible machine $i \in N(j)$ to irrevocably assign the job to before seeing any more jobs in the sequence. We also consider fractional assignments, i.e. vectors belonging to the set $X = \{x \in \mathbb{R}_+^{m \times n} \mid \forall j \in [n], \sum_i x_{ij} = 1, \text{ and } x_{ij} = 0 \iff i \notin N(j)\}$.

Prior work studied the application of proportional weights[3, 26–28]. Intuitively, a prediction in this setting is a weighting of *machines*, which then implies an online assignment, which is shown to be near-optimal. Slightly more formally, suppose that we are given weights $w_i$ for each machine $i$. Then each job $j$ is *fractionally* assigned to machine $i$ to a fractional amount of $\frac{w_i}{\sum_{i' \in N(j)} w_{i'}}$. Notice that given weights, this also gives an online assignment. It is known that there exist weights for any instance where the fractional solution has a near optimal makespan, even though there are only $m$ "degree of freedom" in the weights compared to $mn$ in an assignment. That is, for all machines $i$, $\sum_{j \in [n]} p_j \cdot \frac{w_i}{\sum_{i' \in N(j)} w_{i'}}$ is at most a $(1 + \epsilon)$ factor larger than the optimal makespan for any constant $\epsilon > 0$ [3, 26].

Let $w^*$ be a set of near optimal weights for a given instance. Lattanzi et al. [26] showed the following theorem:

**Theorem 4.1.** *Given predicted weights $\hat{w}$, there is an online fractional algorithm which has makespan $O(\log(\eta(\hat{w}, w^*)\mathrm{OPT})$, where $\eta(\hat{w}, w^*) := \max_{i\in[m]} \max(\frac{\hat{w}_i}{w_i^*}, \frac{w_i^*}{\hat{w}_i})$ to be the error in the prediction.*

Moreover, this fractional assignment can be converted online to an integral assignment while losing only an $O(\log \log m)$ factor in the makespan [26, 28]. Thus, we focus on constructing fractional assignments that are competitive with the best prediction in hindsight.

## 4.2 Combining Fractional Solutions Online

Given $k$ different predicted weight vectors $\hat{w}^1, \hat{w}^2, \ldots, \hat{w}^k$, we want to give an algorithm which is competitive against the *minimum* error among the predicted weights, i.e. we want the competitiveness to depend upon $\eta_{\min} := \min_{\ell \in [k]} \eta(\hat{w}^\ell, w^*)$.

The challenge is that we do not know up front which $\ell \in [k]$ will yield the smallest error, but instead learn this in hindsight. For each $\ell \in [k]$, let $x^\ell$, be the resulting fractional assignment from applying the fractional online algorithm due to [26] with weights $\hat{w}^\ell$. This fractional assignment is revealed one job at a time.

We give an algorithm which is $O(\log k)$-competitive against any collection of $k$ fractional assignments which are revealed online. Moreover, our result applies to the unrelated machines setting, in which each job has a collection of machine-dependent sizes $\{p_{ij}\}_{i \in [m]}$. The algorithm is based on the doubling trick and is similar to results in [10] which apply to metrical task systems. Let $\beta := \min_{\ell \in [k]} \max_i \sum_j p_{ij} x_{ij}^\ell$ be the best fractional makespan in hindsight. As in previous work, our algorithm is assumed to know $\beta$, an assumption that can be removed [26]. At a high level, our algorithm maintains a set $A \subseteq [k]$ of solutions which are good with respect to the current value of $\beta$, averaging among these. See Algorithm 2 for a detailed description. We have the following theorem.

**Theorem 4.2.** *Let $x^1, x^2, \ldots, x^k$ be fractional assignments which are revealed online. If Algorithm 2 is run with $\beta := \min_{\ell \in [k]} \max_i \sum_j p_{ij} x_{ij}^\ell$, then it yields a solution of cost $O(\log k) \cdot \beta$ and never reaches the fail state (line 7 in Algorithm 2).*

Let $\beta_\ell = \max_i \sum_j p_{ij} x_{ij}^\ell$ and OPT be the optimal makespan. Theorem 4.1 shows that $\beta_\ell \leq O(\log \eta_\ell)$OPT. The following corollary is then immediate:

**Corollary 4.3.** *Let $w^1, w^2, \ldots, w^k$ be the predicted weights with errors $\eta^1, \eta^2, \ldots, \eta^k$. Then Algorithm 2 returns a fractional assignment with makespan at most $\text{OPT} \cdot O(\log k) \cdot \min_{\ell \in [k]} \log(\eta^\ell)$.*

---

**Algorithm 2** Algorithm for combining fractional solutions online for load balancing.

---

1: **procedure** COMBINE-LOADBALANCING($\beta$)
2:      $A \leftarrow [k]$                                      ▷ Initially all solutions are good
3:      **for** each job $j$ **do**
4:          Receive the assignments $x^1, x^2, \ldots, x^k$
5:          $A(j, \beta) \leftarrow \{\ell \in A \mid \forall i \in [m], x_{ij}^\ell > 0 \implies p_{ij} x_{ij}^\ell \leq \beta\}$
6:          **if** $A = \emptyset$ or $A(j, \beta) = \emptyset$ **then**
7:              Return "Fail"
8:          **end if**
9:          $\forall i \in [m], x_{ij} \leftarrow \frac{1}{|A(j,\beta)|} \sum_{\ell \in A(j,\beta)} x_{ij}^\ell$
10:         $B \leftarrow \{\ell \in A \mid \max_{i \in [m]} \sum_{j' \leq j} p_{ij'} x_{ij'}^\ell > \beta\}$          ▷ Bad solutions w.r.t. $\beta$
11:         $A \leftarrow A \setminus B$
12:      **end for**
13: **end procedure**

---

We defer the proof of Theorem 4.2 to the Supplementary material.

## 4.3 Learning $k$ Predicted Weight Vectors

We now turn to the question of showing how to learn $k$ different predicted weight vectors $\hat{w}^1, \hat{w}^2, \ldots, \hat{w}^k$. Recall that there is an unknown distribution $\mathcal{D}$ over sets of $n$ jobs from which we receive independent samples $J_1, J_2, \ldots, J_S$. Our goal is to show that we can efficiently learn (in terms of sample complexity) $k$ predicted weight vectors to minimize the expected minimum error. Let $w^*(J)$ be the correct weight vector for instance $J$ and let $\eta(w, w') = \max_{i \in [m]} \max(\frac{w_i}{w_i'}, \frac{w_i'}{w_i})$ be the error between a pair of weight vectors. We have the following result.

**Theorem 4.4.** *Let $\mathcal{D}$ be an unknown distribution over restricted assignment instances on $n$ jobs and let $w^*(J)$ be a set of good weights for instance $J$. Given $S$ independent samples from $\mathcal{D}$,*

*there is a polynomial time algorithm that outputs $k$ weight vectors $\hat{w}^1, \hat{w}^2, \ldots, \hat{w}^k$ such that* $\mathbb{E}_{J \sim \mathcal{D}} \left[ \min_{\ell \in [k]} \log(\eta(\hat{w}^\ell, w^*(J)) \right] \leq O(1) \cdot \min_{w^1, w^2, \ldots, w^k} \mathbb{E} \left[ \min_{\ell \in [k]} \log(\eta(w^\ell, w^*(J)) \right] + \epsilon$ *with probability* $1 - \delta$*, where* $S = \text{poly}(m, k, \frac{1}{\epsilon}, \frac{1}{\delta})$

The proof of Theorem 4.4 is deferred to the Supplementary material, but we note that to get a polynomial time algorithm we carry out an interesting reduction to $k$-median clustering. Namely, we show that the function $d(w, w') := \log(\eta(w, w'))$ satisfies the triangle inequality and thus forms a metric space.

# 5 Scheduling with Predicted Permutations

In this problem there are $n$ jobs, indexed by $1, 2, \ldots, n$, to be scheduled on a single machine. We assume that they are all available at time 0. Job $j$ has size $p_j$ and needs to get processed for $p_j$ time units to complete. If all job sizes are known a priori, Shortest Job First (or equivalently Shortest Remaining Time First), which processes jobs in non-decreasing order of their size, is known to be optimal for minimizing total completion time. We assume that the true value of $p_j$ is unknown and is revealed only when the job completes, i.e. the *non-clairvoyant* setting. In the non-clairvoyant setting, it is known that Round-Robin (which processes all alive jobs equally) is 2-competitive and that this is the best competitive ratio one can hope for [35].

We study this basic scheduling problem assuming certain predictions are available for use. Following the recent work by Lindermayr and Megow [30], we will assume that we are given $k$ orderings/sequences as prediction, $\{\sigma_\ell\}_{\ell \in [k]}$. Each $\sigma_\ell$ is a permutation of $J := [n]$. Intuitively, it suggests an ordering in which jobs should be processed. This prediction is inspired by the aforementioned Shortest Job First (SJF) as an optimal schedule can be described as an ordering of jobs, specifically increasing order of job sizes.

For each $\sigma_\ell$, its error is measured as $\eta(J, \sigma_\ell) := \text{COST}(J, \sigma_\ell) - \text{OPT}(J)$, where $\text{COST}(J, \sigma_\ell)$ denotes the objective of the schedule where jobs are processed in the order of $\sigma_\ell$ and $\text{OPT}(J)$ denotes the optimal objective value. We may drop $J$ from notation when it is clear from the context.

As observed in [30], the error can be expressed as $\eta(J, \sigma_\ell) = \sum_{i < j \in J} I^\ell_{i,j} \cdot |p_i - p_j|$, where $I^\ell_{i,j}$ is an indicator variable for 'inversion' that has value 1 if and only if $\sigma_\ell$ predicts the pairwise ordering of $i$ and $j$ incorrectly. That is, if $p_i < p_j$, then the optimal schedule would process $i$ before $j$; here $I^\ell_{i,j} = 1$ iff $i \succ_{\sigma_\ell} j$.

As discussed in [30], this error measure satisfies two desired properties, monotonicity and Lipschitzness, which were formalized in [24].

Our main result is the following.

**Theorem 5.1.** *Consider a constant $\epsilon > 0$. Suppose that for any $S \subseteq J$ with $|S| = \Theta(\frac{1}{\epsilon^4}(\log \log n + \log k + \log(1/\epsilon)))$, we have $\text{OPT}(S) \leq c\epsilon \cdot \text{OPT}(J)$ for some small absolute constant $c$. Then, there exists a randomized algorithm that yields a schedule whose expected total completion time is at most $(1 + \epsilon)\text{OPT} + (1 + \epsilon)\frac{1}{\epsilon^5}\eta(J, \sigma_\ell)$ for all $\ell \in [k]$.*

As a corollary, by running our algorithm with $1 - \epsilon$ processing speed and simultaneously running Round-Robin with the remaining $\epsilon$ of the speed, the cost increases by a factor of at most $\frac{1}{1-\epsilon}$ while the resulting hybrid algorithm is $2/\epsilon$-competitive.[3]

## 5.1 Algorithm

To make our presentation more transparent we will first round job sizes. Formally, we choose $\rho$ uniformly at random from $[0, 1)$. Then, round up each job $j$'s size to the closest number of the form $(1 + \epsilon)^{\rho + t}$ for some integer $t$. Then, we scale down all job sizes by $(1 + \epsilon)^\rho$ factor. We will present our algorithm and analysis assuming that every job has a size equal to a power of $(1 + \epsilon)$. In the

---

[3]This hybrid algorithm is essentially the preferential time sharing [24, 30, 37]. Formally, we run our algorithm ignoring RR's processing and also run RR ignoring our algorithm; this can be done by a simple simulation. Thus, we construct two schedules concurrently and each job completes at the time when it does in either schedule. This type of algorithms was first used in [37].

supplementary we show how to remove this assumption without increasing our algorithm's objective by more than $1 + \epsilon$ factor in expectation.

We first present the following algorithm that achieves Theorem 5.1 with $|S| = \Theta(\frac{1}{\epsilon^4}(\log n + \log k))$. The improved bound claimed in the theorem needs minor tweaks of the algorithm and analysis and they are deferred to the supplementary material.

Our algorithm runs in rounds. Let $J_r$ be the jobs that complete in round $r \geq 1$. For any subset $S$ of rounds, $J_S := \cup_{r \in S} J_r$. For example, $J_{\leq r} := J_1 \cup \ldots \cup J_r$. Let $n_r := |J_{\geq r}| = n - |J_{<r}|$ denote the number of alive jobs at the beginning of round $r$.

Fix the beginning of round $r$. The algorithm processes the job in the following way for this round. If $n_r \leq \frac{1}{\epsilon^4}(\log n + \log k)$, we run Round-Robin to complete all the remaining jobs, $J_{\geq r}$. This is the last round and it is denoted as round $L + 1$. Otherwise, we do the following Steps 1-4:

**Step 1. Estimating $\epsilon$-percentile.**  Roughly speaking, the goal is to estimate the $\epsilon$-percentile of job sizes among the remaining jobs. For a job $j \in J_{\geq r}$, define its rank among $J_{\geq r}$ as the number of jobs no smaller than $j$ in $J_{\geq r}$ breaking ties in an arbitrary yet fixed way. Ideally, we would like to estimate the size of job of rank $\epsilon n_r$, but do so only approximately.

The algorithm will find $\tilde{q}_r$ that is the size of a job whose rank lies in $[\epsilon(1 - \epsilon)n_r, \epsilon(1 + \epsilon)n_r]$. To handle the case that there are many jobs of the same size $\tilde{q}_r$, we estimate $y_r$ the number of jobs no bigger than $\tilde{q}_r$; let $\tilde{y}_r$ denote our estimate of $y_r$. We will show how we can do these estimations without spending much time by sampling some jobs and partially processing them in Round-Robin manner (the proof of the following lemma can be found in the supplementary material.)

**Lemma 5.2.** *W.h.p. the algorithm can construct estimates $\tilde{q}_r$ and $\tilde{y}_r$ in time at most $O(\tilde{q}_r \frac{1}{\epsilon^2} \log n)$ such that there is a job of size $\tilde{q}_r$ whose rank lies in $[\epsilon(1 - \epsilon)n_r, \epsilon(1 + \epsilon)n_r]$ and $|\tilde{y}_r - y_r| \leq \epsilon^2 n_r$.*

**Step 2. Determining Good and Bad Sequences.**  Let $\sigma_\ell^r$ denote $\sigma_\ell$ with all jobs completed in the previous rounds removed and with the relative ordering of the remaining jobs fixed. Let $\sigma_{\ell,\epsilon}^r$ denote the first $\tilde{y}_r$ jobs in the ordering. We say a job $j$ is big if $p_j > \tilde{q}_r$; middle if $p_j = \tilde{q}_r$; small otherwise. Using sampling and partial processing we will approximately distinguish good and bad sequences. Informally $\sigma_\ell^r$ is good if $\sigma_{\ell,\epsilon}^r$ has few big jobs and bad if it does many big jobs. The proof of the following lemma can be found in the supplementary material.

**Lemma 5.3.** *For all $\ell \in [k]$, we can label sequence $\sigma_\ell^r$ either good or bad in time at most $O(\tilde{q}_r \frac{1}{\epsilon^2}(\log n + \log k))$ that satisfies the following with high probability: If it is good, $\sigma_{\ell,\epsilon}^r$ has at most $3\epsilon^2 n_r$ big jobs; otherwise $\sigma_{\ell,\epsilon}^r$ has at least $\epsilon^2 n_r$ big jobs.*

**Step 3. Job Processing.**  If all sequences are bad, then we process all jobs, each up to $\tilde{q}_r$ units in an arbitrary order. Otherwise, we process the first $\tilde{y}_r$ jobs in an arbitrary good sequence, in an arbitrary order, each up to $\tilde{q}_r$ units.

**Step 4. Updating Sequences.**  The jobs completed in this round drop from the sequences but the remaining jobs' relative ordering remains fixed in each (sub-)sequence. For simplicity, we assume that partially processed jobs were never processed—this is without loss of generality as this assumption only increases our schedule's objective.

## 5.2   Analysis of the Algorithm's Performance

We defer the analysis of the above algorithm (the proof of Theorem 5.1) to the supplementary material, as it is quite technical and complex. At a very high level, though, we use the fact that the error in each prediction can be decomposed into pair-wise inversions, and moreover we can partition the inversions into the rounds of the algorithm in which they appear. Then we look at each round, and split into two cases. First, if all sequences are bad then every prediction has large error, so we can simply use Round Robin (which is 2-competitive against OPT) and the cost can be charged to the error of any prediction. Second, if there is a good sequence, then in any good sequence the number of big jobs is small (so we do not spend much time processing them), and we therefore complete almost all of the non-big jobs. Here, we crucially use the fact that we can process the first $\epsilon$ fraction of jobs in a sequence in an arbitrary order remaining competitive against the sequence. Finally, we show that all

of the additional assumptions and costs (e.g., rounding processing times and the cost due to sampling) only change our performance by a $1 + \epsilon$ factor. Getting all of these details right requires much care.

### 5.3 Learning $k$ Predicted Permutations

Now we show that learning the best $k$ permutations has polynomial sample complexity.

**Theorem 5.4.** *Let $\mathcal{D}$ be an unknown distribution of instances on $n$ jobs. Given $S$ independent samples from $\mathcal{D}$, there is an algorithm that outputs $k$ permutations $\hat{\sigma}_1, \hat{\sigma}_2, \ldots, \hat{\sigma}_k$ such that $\mathbb{E}_{J \sim \mathcal{D}} \left[ \min_{\ell \in [k]} \eta(J, \hat{\sigma}_\ell) \right] \leq \min_{\sigma_1, \sigma_2, \ldots, \sigma_k} \mathbb{E}_{J \sim \mathcal{D}} \left[ \min_{\ell \in [k]} \eta(J, \sigma_\ell) \right] + \epsilon$ with probability $1 - \delta$, where $S = \mathrm{poly}(n, k, \frac{1}{\epsilon}, \frac{1}{\delta})$.*

*Proof.* The algorithm is basic ERM, and the polynomial sample complexity follows from Theorem 2.1 and Theorem 20 in Lindermayr and Megow [30]. $\square$

## 6   Conclusion

Despite the explosive recent work in algorithms with predictions, almost all of this work has assumed only a single prediction. In this paper we study algorithms with *multiple* machine-learned predictions, rather than just one. We study three different problems that have been well-studied in the single prediction setting but not with multiple predictions: faster algorithms for min-cost bipartite matching using learned duals, online load balancing with learned machine weights, and non-clairvoyant scheduling with order predictions. For all of the problems we design algorithms that can utilize multiple predictions, and show sample complexity bounds for learning the best set of $k$ predictions. Demonstrating the effectiveness of our algorithms (and the broader use of multiple predictions) empirically is an interesting direction for further work.

Surprisingly, we have shown that in some cases, using multiple predictions is essentially "free." For instance, in the case of min-cost perfect matching examining $k = O(\sqrt{n})$ predictions takes the same amount of time as one round of the Hungarian algorithm, but the number of rounds is determined by the quality of the *best* prediction. In contrast, for load balancing, using $k$ predictions always incurs an $O(\log k)$ cost, so using a constant number of predictions may be best. More generally, studying this trade-off between the cost and the benefit of multiple predictions for other problems remains an interesting and challenging open problem.

### Acknowledgments and Disclosure of Funding

Michael Dinitz was supported in part by NSF grant CCF-1909111. Sungjin Im was supported in part by NSF grants CCF-1617653, CCF-1844939 and CCF-2121745. Thomas Lavastida and Benjamin Moseley were supported in part by NSF grants CCF-1824303, CCF-1845146, CCF-2121744 and CMMI-1938909. Benjamin Moseley was additionally supported in part by a Google Research Award, an Infor Research Award, and a Carnegie Bosch Junior Faculty Chair.

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
