In this section we discuss the learnability of $k$ predictions and its connection to $k$-median clustering, which we apply to specific problems in later sections.

**Learnability and Pseudo-dimension:** Consider a problem $\mathcal{I}$ and let $\mathcal{D}$ be an unknown distribution over instances of $\mathcal{I}$. We are interested in the learnability of predictions for such problems with respect to an error function. Suppose that the predictions come from some space $\Theta$ and for a given instance $I \in \mathcal{I}$ and prediction $\theta \in \Theta$ the error is $\eta(I, \theta)$. Given $S$ independent samples $\{I_s\}_{s=1}^{S}$ from $\mathcal{D}$, we would like to compute $\hat{\theta} \in \Theta$ such that with probability at least $1 - \delta$, we have that

$$\mathbb{E}_{I \sim \mathcal{D}}[\eta(I, \hat{\theta})] \leq \min_{\theta \in \Theta} \mathbb{E}_{I \sim \mathcal{D}}[\eta(I, \theta)] + \epsilon. \tag{1}$$

We would like $S$ (the sample complexity) to be polynomial in $\frac{1}{\epsilon}, \frac{1}{\delta}$ and other parameters of the problem $\mathcal{I}$ (e.g. the number of vertices in matching, or the number of jobs and machines in scheduling and load balancing).

The natural algorithm for solving this problem is empirical risk minimization (ERM): take $\hat{\theta} = \arg\min_{\theta \in \Theta} \frac{1}{S} \sum_{s=1}^{S} \eta(I_s, \theta)$. The sample complexity of ERM, i.e. how large $S$ should be so that (1) holds, can be understood in terms of the pseudo-dimension of the class of functions $\{\eta(\cdot, \theta) \mid \theta \in \Theta\}$. More generally, the pseudo-dimension can be defined for any class of real valued function on some space $\mathcal{X}$.

**Definition 2.1.** *[8, 36, 38] Let $\mathcal{F}$ be a class of functions $f : \mathcal{X} \rightarrow \mathbb{R}$. Let $C = \{x_1, x_2, \ldots, x_S\} \subset \mathcal{X}$. We say that that $C$ is* shattered *by $\mathcal{F}$ if there exist real numbers $r_1, \ldots, r_S$ so that for all $C' \subseteq C$, there is a function $f \in \mathcal{F}$ such that $f(x_i) \leq r_i \iff x_i \in C'$ for all $i \in [S]$. The* pseudo-dimension *of $\mathcal{F}$ is the largest $S$ such that there exists an $C \subseteq X$ with $|C| = S$ that is shattered by $\mathcal{F}$.*

The pseudo-dimension allows us to give a bound on the number of samples required for uniform convergence, which in turn can be used to show that ERM is sufficient for achieving (1).

**Theorem 2.2.** *[8, 36, 38] Let $\mathcal{D}$ be a distribution over a domain $X$ and $\mathcal{F}$ be a class of functions $f : \mathcal{X} \rightarrow [0, H]$ with pseudo-dimension $d_{\mathcal{F}}$. Consider $S$ independent samples $x_1, x_2, \ldots, x_S$ from $\mathcal{D}$. There is a universal constant $c_0$, such that for any $\epsilon > 0$ and $\delta \in (0, 1)$, if $S \geq c_0 \left(\frac{H}{\epsilon}\right)^2 (d_{\mathcal{F}} + \ln(1/\delta))$ then we have*

$$\left| \frac{1}{s} \sum_{s=1}^{S} f(x_i) - \mathbb{E}_{x \sim \mathcal{D}}[f(x)] \right| \leq \epsilon$$

*for all $f \in \mathcal{F}$ with probability at least $1 - \delta$.*

We can extend this learning problem to the setting of multiple predictions. The setup is the same as above, except that now we are interested in outputting $k$ predictions $\hat{\theta}^1, \hat{\theta}^2, \ldots, \hat{\theta}^k$ such that with probability at least $1 - \delta$:

$$\mathbb{E}_{I \sim \mathcal{D}} \left[ \min_{\ell \in [k]} \eta(I, \hat{\theta}^\ell) \right] \leq \min_{\theta^1, \theta^2, \ldots, \theta^k \in \Theta} \mathbb{E}_{I \sim \mathcal{D}} \left[ \min_{\ell \in [k]} \eta(I, \theta^\ell) \right] + \epsilon. \tag{2}$$

We can again consider ERM algorithms for this task, and we would like to bound the sample complexity. We do this by showing that if the pseudo-dimension of the class of functions associated with one prediction is bounded, then it is also bounded for $k$ predictions. More formally, we want to bound the pseudo-dimension of the class of functions $\{\min_{\ell \in [k]} \eta(\cdot, \theta^\ell) \mid \theta^1, \theta^2, \dots, \theta^k \in \Theta\}$. This can be done via the following result combined with Theorem 2.2 (assuming that the pseudo-dimension of $\{\eta(\cdot, \theta) \mid \theta \in \Theta\}$ is bounded).

**Theorem 2.3.** *Let $\mathcal{F}$ be a class of functions $f : \mathcal{X} \to \mathbb{R}$ with pseudo-dimension $d$ and let $\mathcal{F}^k := \{F(x) = \min_{\ell \in [k]} f^\ell(x) \mid f^1, f^2, \dots, f^k \in \mathcal{F}\}$. Then the pseudo-dimension of $\mathcal{F}^k$ is at most $\tilde{O}(dk)$.*

*Proof.* To show this we first relate things back to VC-dimension.

**Proposition 2.4.** *Let $\mathcal{F}$ be a class of real valued functions on $\mathcal{X}$. Define $\mathcal{H}$ as a class of binary functions on $\mathcal{X} \times \mathbb{R}$ as $\mathcal{H} = \{h(x, r) = \text{sgn}(f(x) - r) \mid f \in \mathcal{F}\}$. Then the pseudo-dimension of $\mathcal{F}$ equals the VC dimension of $\mathcal{H}$.*

The above proposition follows directly from the definition of pseudo- and VC-dimensions. The next lemma we need is well known.

**Proposition 2.5** (Sauer-Shelah Lemma)**.** *If $\mathcal{H}$ has VC-dimension $d$ and $x_1, \dots, x_m$ is a sample of size $m$, then the number of sets shattered by $\mathcal{H}$ is at most $O(m^d)$.*

Let $x_1, \dots, x_m \in \mathcal{X}$ and $r_1, \dots, r_m \in \mathbb{R}$ be given. We upper bound the number of possible labelings induced by $\mathcal{F}_k$ on this set. Note that on this sample the Sauer-Shelah Lemma implies that the number of labelings induced by $\mathcal{F}$ on this sample is at most $O(m^d)$. $\mathcal{F}_k$ allows us to choose $k$ functions from $\mathcal{F}$ so this increases the number of possible labelings to at most $O(m^{dk})$. We shatter this set if this bound is greater than $2^m$. Reorganizing these bounds implies that $m = \tilde{O}(dk)$, which implies the upper bound on the pseudo-dimension of $\mathcal{F}_k$. $\qquad\qquad\square$

The associated ERM problem for computing $k$ predictions becomes more interesting. Recall that in this problem we are given a sample of $S$ instances $I_1, I_2, \dots, I_S \sim \mathcal{D}$ and we want to compute $\hat{\theta}^1, \hat{\theta}^2, \dots, \hat{\theta}^k \in \Theta$ in order to minimize $\frac{1}{S} \sum_{s=1}^{S} \min_{\ell \in [k]} \eta(I_s, \hat{\theta}^\ell)$. This can be seen as an instance of the $k$-median clustering problem where we want to cluster the "points" $\{I_s\}_{s=1}^{S}$ by opening $k$ "facilities" from the set $\Theta$ and the cost of assigning $I_s$ to $\theta$ is $\eta(I_s, \theta)$. In general, this problem may be hard to solve or even approximate. In the case that the costs have some metric structure, then it is known how to compute $O(1)$-approximate solutions [29]. For minimum cost matching (Section 3 and load balancing (Section 4), we will show that the ERM problem can be seen as a $k$-median problem on an appropriate (pseudo-)metric space.

**Metrics and Clustering:** Recall that $(\mathcal{X}, d)$ is a metric space if the distance function $d : \mathcal{X} \times \mathcal{X} \to \mathbb{R}_+$ satisfies the following properties:

1. For all $x, y \in \mathcal{X}$, $d(x, y) = 0 \iff x = y$
2. For all $x, y \in \mathcal{X}$, $d(x, y) = d(y, x)$
3. For all $x, y, z \in \mathcal{X}$, $d(x, z) \leq d(x, y) + d(y, z)$

If we replace the first property with the weaker property that for all $x \in \mathcal{X}, d(x, x) = 0$, then we call $(\mathcal{X}, d)$ a pseudo-metric space.

Given a finite set of points $X \subseteq \mathcal{X}$, the $k$-median clustering problem is to choose a subset $C \subseteq \mathcal{X}$ of $k$ centers to minimize the total distance of each point in $X$ to its closest center in $C$. In notation, the goal of $k$-median clustering is to solve

$$\min_{C \subseteq \mathcal{X}, |C|=k} \sum_{x \in X} \min_{c \in C} d(x, c) \tag{3}$$

In our settings it will often be challenging to optimize $C$ over all of $\mathcal{X}$, so at an $O(1)$-factor loss to the objective we can instead optimize $C$ over $X$. Formally, we have the following standard lemma.

**Lemma 2.6.** *Let $(\mathcal{X}, d)$ be a pseudo-metric space and let $X$ be a finite subset of $\mathcal{X}$. Then for all $k > 0$ we have*

$$\min_{C \subseteq X, |C|=k} \sum_{x \in X} \min_{c \in C} d(x,c) \leq 2 \cdot \min_{C \subseteq \mathcal{X}, |C|=k} \sum_{x \in X} \min_{c \in C} d(x,c).$$

*Proof.* Let $C^* \subseteq \mathcal{X}$ be an optimal solution to the problem on the right hand side of the inequality, and let its cost be OPT. We consider a mapping $\phi : C^* \to X$, which gives us a solution to the problem on the left hand side by taking $C = \{\phi(c) \mid c \in C^*\}^2$. For $c \in C^*$, define $\phi(c) = \arg\min_{x \in X} d(x,c)$ We will argue that the cost of $C$ is at most 2OPT. Let $x \in X$ and let $c^* = \arg\min_{c \in C^*} d(x,c)$ be its closest center in $C^*$, then we have

$$\min_{c \in C} d(x,c) \leq d(x, \phi(c^*)) \leq d(x, c^*) + d(c^*, \phi(c^*)) \leq 2d(x, c^*) = 2 \min_{c \in C^*} d(x,c)$$

The second to last inequality follows from the triangle inequality and the last follows from the definition of $\phi$. Now summing over all $x \in X$ yields that the cost of using $C$ is at most 2OPT. $\qquad\square$

## 3 Minimum Cost Bipartite Matching with Predicted Duals

In this section we study the minimum cost bipartite matching problem with multiple predictions. The case of a single prediction has been considered recently [17, 18], where they used dual values as a prediction and showed that the classical Hungarian algorithm could be sped up by using appropriately learned dual values. Our goal in this section is to extend these results to multiple predictions, i.e., multiple duals. In particular, in Section 3.2 we show that we can use $k$ duals and get running time comparable to the time we would have spent if we used the single best of them in the algorithm of [18], with no asymptotic loss if $k$ is at most $O(\sqrt{n})$. Then in Section 3.3 we show that $k$ predictions can be learned with not too many more samples (or running time) than learning a single prediction.

### 3.1 Problem Definition and Predicted Dual Variables

In the minimum cost bipartite matching problem we are given a bipartite graph $G = (V, E)$ with $n = |V|$ vertices and $m = |E|$ edges, with edge costs $c \in \mathbb{Z}^E$. The objective is to output a perfect matching $M \subseteq E$ which minimizes the cost $c(M) := \sum_{e \in E} c_e$. This problem is exactly captured by the following primal and dual linear programming formulations.

$$
\begin{aligned}
\min \quad & \sum_{e \in E} c_e x_e \\
& \sum_{e \in N(i)} x_e = 1 \quad \forall i \in V \qquad\qquad \text{(MWPM-P)} \\
& \qquad x_e \geq 0 \qquad \forall e \in E
\end{aligned}
$$

$$
\begin{aligned}
\max \quad & \sum_{i \in V} y_i \\
& y_i + y_j \leq c_e \quad \forall e = ij \in E \qquad \text{(MWPM-D)}
\end{aligned}
$$

Dinitz et al. [18] studied initializing the Hungarian algorithm with a prediction $\hat{y}$ of the optimal dual solution $y^*$. They propose an algorithm which operates in two steps (see Algorithm 1 for pseudo-code). First, the predicted dual solution $\hat{y}$ may not be feasible, so they give an $O(n + m)$ time algorithm which recovers feasibility (which we refer to as Make-Feasible). Second, the now-feasible dual solution is used in a primal-dual algorithm such as the Hungarian algorithm (which we refer to as Primal-Dual) and they show that the running time depends on the $\ell_1$ error in the predicted solution. In addition to this they show that learning a good initial dual solution is computationally efficient with low sample complexity. More formally, they proved the following theorems.

**Theorem 3.1** (Dinitz et al. [18]). *Let $(G, c)$ be an instance of minimum cost bipartite matching and $\hat{y}$ be a prediction of an optimal dual solution $y^*$. Algorithm 1 returns an optimal solution and runs in time $O(m\sqrt{n} \cdot \|y^* - \hat{y}\|_1)$. Moreover, the Make-Feasible step of Algorithm 1 runs in $O(n + m)$ time.*

---

[2]In the case that $|C| < k$, adding arbitrary points from $X$ to $C$ so that $|C| = k$ can only decrease the cost from just using $C$.

**Theorem 3.2** (Dinitz et al. [18]). *Let $\mathcal{D}$ be an unknown distribution over instances $(G, c)$ on $n$ vertices and let $y^*(G, c)$ be an optimal dual solution for the given instance. Given $S$ independent samples from $\mathcal{D}$, there is a polynomial time algorithm that outputs a solution $\hat{y}$ such that*

$$\mathbb{E}_{(G,c)\sim\mathcal{D}}\left[\|y^*(G, c) - \hat{y}\|_1\right] \leq \min_y \mathbb{E}_{(G,c)\sim\mathcal{D}}\left[\|y^*(G, c) - y\|_1\right] + \epsilon$$

*with probability $1 - \delta$ where $S = \mathrm{poly}(n, \frac{1}{\epsilon}, \frac{1}{\delta})$.*

---

**Algorithm 1** Minimum cost matching with a predicted dual solution

1: **procedure** PREDICTEDPRIMAL-DUAL$(G, c, \hat{y})$
2: $\quad$ $y \leftarrow$ MakeFeasible$(G, c, \hat{y})$
3: $\quad$ $M \leftarrow$ Primal-Dual$(G, c, y)$
4: $\quad$ Return $M$
5: **end procedure**

---

### 3.2 Using $k$ Predicted Dual Solutions Efficiently

Given $k$ predicted dual solutions $\hat{y}^1, \hat{y}^2, \ldots, \hat{y}^k$, we would like to efficiently determine which solution has the minimum error for the given problem instance. Note that the predicted solutions may still be infeasible and that we do not know the target optimal dual solution $y^*$. We propose the following simple algorithm which takes as input $k$ predicted solutions and whose running time depends only on the $\ell_1$ error of the *best* predicted solution. First, we make each predicted solution feasible, just as before. Next, we select the (now-feasible) dual solution with highest dual objective value and proceed running the primal-dual algorithm with only that solution. See Algorithm 2 for pseudo-code.

---

**Algorithm 2** Minimum cost matching with $k$ predicted dual solutions

1: **procedure** $k$-PREDICTEDPRIMAL-DUAL$(G, c, \hat{y}^1, \hat{y}^2, \ldots, \hat{y}^k)$
2: $\quad$ **for** $\ell \in [k]$ **do**
3: $\quad\quad$ $y^\ell \leftarrow$ MakeFeasible$(G, c, \hat{y}^\ell)$
4: $\quad$ **end for**
5: $\quad$ $\ell' \leftarrow \arg\max_{\ell \in [k]} \sum_{i \in V} y_i^\ell$
6: $\quad$ $M \leftarrow$ Primal-Dual$(G, c, y^{\ell'})$
7: $\quad$ Return $M$
8: **end procedure**

---

We have the following result concerning Algorithm 2. To interpret this result, note that the cost for increasing the number of predictions is $O(k(n + m))$, which will be dominated by the $m\sqrt{n}$ term we pay for running the Hungarian algorithm unless $k$ is extremely large (certainly larger than $\sqrt{n}$) or there is a prediction with 0 error (which is highly unlikely). Hence we can reap the benefit of a large number of predictions "for free".

**Theorem 3.3.** *Let $(G, c)$ be a minimum cost bipartite matching instance and let $\hat{y}^1, \hat{y}^2, \ldots, \hat{y}^k$ be predicted dual solutions. Algorithm 2 returns an optimal solution and runs in time $O(k(n + m) + m\sqrt{n} \cdot \min_{\ell \in [k]} \|y^* - \hat{y}^\ell\|_1)$.*

*Proof.* The correctness of the algorithm (i.e., returning an optimal solution) follows from the correctness of Algorithm 1. For the running time, we clearly spend $O(k(n + m))$ time making each predicted solution feasible, thus we just need to show the validity of latter term in the running time. Let $\ell' = \arg\max_{\ell \in [k]} \sum_{i \in V} y_i^\ell$ be the solution chosen in line 5 of Algorithm 2 and let $\ell^* = \arg\min_{\ell \in [k]} \|y^* - \hat{y}^\ell\|_1$ be the solution with minimum error. Recall that for each $\ell \in [k]$, $y^\ell$ is the resulting *feasible* dual solution from calling Make-Feasible on $\hat{y}^\ell$. By the analysis from [18], we have that $\|y^* - y^{\ell^*}\|_1 \leq 3\|y^* - \hat{y}^{\ell^*}\|_1$, so it suffices to show that the number of primal-dual iterations will be bounded by $\|y^* - y^{\ell^*}\|_1$. By our choice of $\ell'$, we have $\sum_i y_i^{\ell'} \geq \sum_i y_i^{\ell^*}$, therefore we have that $\sum_i (y_i^* - y^{\ell'}) \leq \sum_i (y_i^* - y_i^{\ell^*}) \leq \|y^* - y^{\ell^*}\|_1$. From the analysis in [18], we have that the number of primal-dual iterations will be at most $\sum_i (y_i^* - y^{\ell'})$, completing the proof. $\qquad\square$

### 3.3 Learning $k$ Predicted Dual Solutions

Next we extend Theorem 3.2 to the setting where we output $k$ predictions. Let $\mathcal{D}$ be a distribution over problem instances $(G, c)$ on $n$ vertices. We show that we can find the best set of $k$ predictions. More formally, we prove the following theorem.

**Theorem 3.4.** *Let $\mathcal{D}$ be an unknown distribution over instances $(G, c)$ on $n$ vertices and let $y^*(G, c)$ be an optimal dual solution for the given instance. Given $S$ independent samples from $\mathcal{D}$, there is a polynomial time algorithm that outputs $k$ solutions $\hat{y}^1, \hat{y}^2, \ldots, \hat{y}^k$ such that*

$$\mathbb{E}_{(G,c)\sim\mathcal{D}}\left[\min_{\ell\in[k]}\|y^*(G,c)-\hat{y}^\ell\|_1\right] \leq O(1)\cdot\min_{y^1,y^2,\ldots,y^k}\mathbb{E}_{(G,c)\sim\mathcal{D}}\left[\min_{\ell\in[k]}\|y^*(G,c)-y^\ell\|_1\right] + \epsilon$$

*with probability $1 - \delta$ where $S = \mathrm{poly}(n, k, \frac{1}{\epsilon}, \frac{1}{\delta})$.*

*Proof.* By Theorem 7 in [18] and Theorems 2.2 and 2.3, we get the polynomial sample complexity. Thus we just need to give a polynomial time ERM algorithm to complete the proof. Let $\{(G^s, c^s)\}_{s=1}^S$ be the set of sampled instances. We start by computing $z^s = y^*(G^s, c^s) \in \mathbb{Z}^V$ for each $s \in [S]$. Consider the ERM problem where we replace the expectation by a sample average:

$$\min_{y^1,y^2,\ldots,y^k}\frac{1}{S}\sum_{s=1}^S\min_{\ell\in[k]}\|z^s - y^\ell\|_1$$

This can be seen as a $k$-median clustering problem where each predicted solution $y^\ell$ is a cluster center and distances are given by the $\ell_1$ norm. Thus we can find an $O(1)$-approximate solution $\hat{y}^1, \hat{y}^2, \ldots, \hat{y}^k$ to this problem which is of polynomial size in polynomial time (for example by applying the algorithm due to [29]). $\qquad\square$

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

---

The analysis relies on the following decomposition of machine $i$'s load. Note that in our algorithm we assign a weight $\alpha^\ell_j \in [0, 1]$ to solution $\ell$ when computing the fractional assignment for job $j$. That is, we take $x_{ij} = \sum_{\ell \in A} \alpha^\ell_j x^\ell_{ij}$ where $\alpha^\ell_j = 1/|A(j, \beta)|$ if $\ell \in A(j, \beta)$ and 0 otherwise. Then the decomposition of machine $i$'s load $L_i$ is

$$L_i = \sum_j p_{ij} x_{ij} = \sum_j p_{ij} \sum_{\ell \in [k]} \alpha^\ell_j x^\ell_{ij} = \sum_{\ell \in [k]} \left( \sum_j \alpha^\ell_j p_{ij} x^\ell_{ij} \right). \tag{4}$$

Now we define $C_i^\ell := \sum_j \alpha_j^\ell p_{ij} x_{ij}^\ell$ to be the contribution of solution $\ell$ to machine $i$'s load. Without loss of generality suppose that the order in which solutions are removed from $S$ in Algorithm 3 is $1, 2, \ldots, k$.

**Lemma 4.4.** *For all $i \in [m]$ and each $\ell \in [k]$, we have that $C_i^\ell \le \frac{2\beta}{k-\ell+1}$.*

*Proof.* Let $j$ be the job in which $\ell$ is removed from $A$ by our algorithm. Before this, for all $j' < j$ we had that $\ell$'s fractional makespan was at most $\beta$. Thus we can write $C_i^\ell$ as:

$$C_i^\ell = \sum_j \alpha_j^\ell p_{ij} x_{ij}^\ell = \sum_{j'<j} \alpha_{j'}^\ell p_{ij'} x_{ij'}^\ell + \alpha_j^\ell p_{ij} x_{ij}^\ell + \sum_{j'>j} \alpha_{j'}^\ell p_{ij'} x_{ij'}^\ell$$

For the first set of terms, we have that $\alpha_{j'}^\ell \le \frac{1}{k-\ell+1}$ since $\ell$ has not yet been removed from $A$. Thus these terms can be bounded above by $\frac{\beta}{k-\ell+1}$, since $\ell$'s fractional makespan was bounded above by $\beta$ before job $j$. For the middle term, we use a similar observation for $\alpha_j^\ell$ but also apply the definition of $S(j,\beta)$ to conclude that $\alpha_j^\ell > 0 \implies p_{ij} x_{ij}^\ell \le \beta$. Thus we get a bound of $\frac{\beta}{k-\ell+1}$ for the middle term. For the final set of terms, we have $\alpha_{j'}^\ell = 0$ for all $j' > j$ since we have removed $\ell$ from $A$ at this point, and so these contribute nothing. Combining these bounds gives the lemma. □

*Proof of Theorem 4.2.* By (4), we have that the load of machine $i$ is at most $\sum_{\ell \in [k]} C_i^\ell$. Applying Lemma 4.4 we have $C_i^\ell \le \frac{2\beta}{k-\ell+1}$. Thus machine $i$'s load is at most $\sum_{\ell \in [k]} \frac{2\beta}{k-\ell+1} = 2H_k\beta = O(\log k) \cdot \beta$, where $H_k = \sum_{\ell=1}^k \frac{1}{k}$ is the $k$'th harmonic number. Now if we run the algorithm with $\beta = \min_{\ell \in [k]} \max_i \sum_j p_{ij} x_{ij}^\ell$, then there is some solution $\ell^* \in [k]$ which has a fractional makespan of $\beta$, so it never gets removed from $A$ or $A(j,\beta)$ in Algorithm 3. In this case Algorithm 3 never fails, completing the proof of the theorem. □

### 4.3 Learning $k$ Predicted Weight Vectors

We now turn to the question of showing how to learn $k$ different predicted weight vectors $\hat{w}^1, \hat{w}^2, \ldots, \hat{w}^k$. Recall that there is an unknown distribution $\mathcal{D}$ over sets of $n$ jobs from which we receive independent samples $J_1, J_2, \ldots, J_S$. Our goal is to show that we can efficiently learn (in terms of sample complexity) $k$ predicted weight vectors to minimize the expected minimum error. Let $w^*(J)$ be the correct weight vector for instance $J$ and let $\eta(w, w') = \max_{i \in [m]} \max(\frac{w_i}{w_i'}, \frac{w_i'}{w_i})$ be the error between a pair of weight vectors. We have the following result.

**Theorem 4.5.** *Let $\mathcal{D}$ be an unknown distribution over restricted assignment instances on $n$ jobs and let $w^*(J)$ be a set of good weights for instance $J$. Given $S$ independent samples from $\mathcal{D}$, there is a polynomial time algorithm that outputs $k$ weight vectors $\hat{w}^1, \hat{w}^2, \ldots, \hat{w}^k$ such that $\mathbb{E}_{J \sim \mathcal{D}} \left[ \min_{\ell \in [k]} \log(\eta(\hat{w}^\ell, w^*(J)) \right] \le O(1) \cdot \min_{w^1, w^2, \ldots, w^k} \mathbb{E} \left[ \min_{\ell \in [k]} \log(\eta(w^\ell, w^*(J)) \right] + \epsilon$ with probability $1 - \delta$, where $S = \mathrm{poly}(m, k, \frac{1}{\epsilon}, \frac{1}{\delta})$*

Prior work [28] has observed that we can take the weights to be from the set $\mathcal{W}(R) = \{w \in \mathbb{R}_+^m \mid \forall i \in [m], \exists \alpha \in [R], \text{ s.t. } w_i = (1 + \epsilon/m)^\alpha\}$. Moreover, it suffices to take $R = \Theta(m^2 \log(m))$ in order to guarantee that for any instance there exists some set of weights in $\mathcal{W}(R)$ such that the associated fractional assignment yields an $O(1)$-approximate solution. Since this set is finite, with only $m^{O(m^2 \log m)}$ members, it follows that the pseudo-dimension of any class of functions parameterized by the weights is bounded by $\log(|\mathcal{W}(R)|) = O(m^2 \log m)$, and so we get polynomial sample complexity. Thus in order to prove Theorem 4.5, it suffices to give a polynomial time ERM algorithm for this problem. As hinted at in the statement of Theorem 4.5, we will be working with the logarithms of the errors. Working in this space allows us to carry out a reduction to $k$-median clustering.

For any pair of weight vectors $w, w' \in \mathbb{R}_+^m$, recall that we define the error between them to be $\eta(w, w') = \max_{i \in [m]} \max(\frac{w_i}{w_i'}, \frac{w_i'}{w_i})$. Note that $\eta(w, w') \ge 1$ with equality if and only if $w = w'$ and that $\eta(w, w') = \eta(w', w)$. The main lemma is that defining $d(w, w') := \log(\eta(w, w'))$ satisfies the triangle inequality, and thus forms a metric on $\mathbb{R}_+^m$ due to the aforementioned observations.

**Lemma 4.6.** *Let $\eta : \mathbb{R}_+^m \times \mathbb{R}_+^m$ and $d : \mathbb{R}_+^m \times \mathbb{R}_+^m$ be defined as above. Then $(\mathbb{R}_+^m, d)$ forms a metric space.*

*Proof.* It is easy to see that $d(w, w') \geq 0$ with equality if and only if $w = w'$ and that $d(w, w') = d(w', w)$. Thus we just need to show that the triangle inequality holds, i.e. that for all $w, w', w''$ we have $d(w, w'') \leq d(w, w') + d(w', w'')$. This will hold as a result of the following claim. For all $w, w', w''$. we have:

$$\eta(w, w'') \leq \eta(w, w') \cdot \eta(w', w'').  \tag{5}$$

Now the triangle inequality follows since

$$d(w, w') = \log(\eta(w, w'')) \leq \log(\eta(w, w')) + \log(\eta(w', w'')) = d(w, w') + d(w', w'').$$

To prove the claim we have the following:

$$
\begin{aligned}
\eta(w, w'') &= \max_i \left\{ \max \left( \frac{w_i}{w_i''}, \frac{w_i''}{w_i} \right) \right\} \\
&= \max_i \left\{ \max \left( \frac{w_i}{w_i'} \frac{w_i'}{w_i''}, \frac{w_i''}{w_i'} \frac{w_i'}{w_i} \right) \right\} \\
&\leq \max_i \left\{ \max \left( \frac{w_i}{w_i'}, \frac{w_i'}{w_i''} \right) \cdot \max \left( \frac{w_i''}{w_i'}, \frac{w_i'}{w_i} \right) \right\} \\
&\leq \max_i \left\{ \max \left( \frac{w_i}{w_i'}, \frac{w_i'}{w_i''} \right) \right\} \cdot \max_i \left\{ \max \left( \frac{w_i''}{w_i'}, \frac{w_i'}{w_i} \right) \right\} \\
&= \eta(w, w') \cdot \eta(w', w'').
\end{aligned}
$$

The two inequalities above follow from the next two claims below. $\qquad\square$

**Claim 4.7.** *For all $a, b, c > 0$ we have $\max(\frac{a}{c}, \frac{c}{a}) \leq \max(\frac{a}{b}, \frac{b}{a}) \cdot \max(\frac{b}{c}, \frac{c}{b})$*

*Proof.* Without loss of generality, we may assume that $a \geq c$. Now we have several cases depending on the value of $b$. For the first case, lets consider when $a \geq b \geq c > 0$. In this case, the left hand side evaluates to $\frac{a}{c}$ while the right hand side also evaluates to $\frac{a}{c}$, so the inequality is valid in this case.

For the next case, consider when $b \geq a \geq c > 0$. In this case, the left hand side is still $\frac{a}{c}$, while the right hand side evaluates to $\frac{b^2}{ac} \geq \frac{ab}{ac} \geq \frac{a}{c}$. Thus the inequality is valid.

In the final case, we have $a \geq c \geq b > 0$. The left hand side is $\frac{a}{c}$, while the right hand side evaluates to $\frac{ac}{b^2} \geq \frac{ac}{c^2} = \frac{a}{c}$, completing the proof. $\qquad\square$

**Claim 4.8.** *Let $u, v \in \mathbb{R}_+^m$, then we have $\max_i(u_i v_i) \leq (\max_i u_i)(\max_i v_i)$*

*Proof.* Suppose for contradiction that this isn't the case, i.e. $\max_i(u_i v_i) > (\max_i u_i)(\max_i v_i)$. Now let $i^* = \arg\max_i(u_i v_i)$. Then we have

$$u_{i^*} v_{i^*} > (\max_i u_i)(\max_i v_i) \geq u_{i^*} v_{i^*}$$

which is the desired contradiction. $\qquad\square$

*Proof of Theorem 4.5.* The sample complexity follows from the discussion above and Theorem 2.3 which shows that the pseudo-dimension of the error function is at most $O(m^2 \log(m))$. Given $S$ samples $J_1, J_2, \ldots, J_S$ from $\mathcal{D}$, we want to solve the corresponding ERM instance. To do this we set up the following $k$-median instance to compute the predicted weights $\hat{w}^1, \hat{w}^2, \ldots, \hat{w}^k$. First we compute $w^s = w^*(J_s)$ for each $s \in [S]$, then we set the distance between $w^s$ and $w^{s'}$ to be the distance function $d(w^s, w^{s'})$ defined above. At a loss of a factor of 2, we can take each $\hat{w}^\ell$ to be in $\{w^s\}_{s=1}^S$ by Lemma 2.6. Thus we can apply an $O(1)$-approximate $k$-median algorithm (e.g. the one due to [29]) to get the

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

### 5.1.1 Subprocedure: Step 1

We detail the first step of the algorithm. Our goal is to prove Lemma 5.2. To obtain the desired estimates, we take a sample $S_r$ of size $\frac{1}{\epsilon^2} \log n$ from $J_{\geq r}$ with replacement. The algorithm processes the sampled jobs using Round-Robin until we complete $\epsilon|S_r|$ jobs.[4] Note that there could be multiple jobs that complete at the same time. This is particularly possible because we assumed that jobs have sizes equal to a power of $1 + \epsilon$. In this case, we break ties in an arbitrary but fixed order. Let $\tilde{q}_r$ be the size of the job that completes $\epsilon|S_r|$th. If $m_r$ is the number of jobs in $S_r$ we completed, we estimate $\tilde{y}_r = \frac{m_r}{|S_r|} n_r$.

Let $B_1$ is the bad event that $\tilde{q}_r$ has a rank that doesn't belong to $[a_1 := \epsilon(1 - \epsilon)n_r, a_2 := \epsilon(1 + \epsilon)n_r]$. Let $X_1$ be the number of jobs sampled that have rank at most $a_1$. Similarly let $X_2$ be the number of jobs sampled that have rank at most $a_2$. Note that $\neg B_1$ occurs if $X_1 \leq \epsilon|S_r| \leq X_2$. Thus, we have $\Pr[B_1] \leq \Pr[X_1 > \epsilon|S_r|] + \Pr[X_2 < \epsilon|S_r|]$. Note that $X_1 = Binomial(a_1/n_r, |S_r|)$.

We use the following well-known Hoeffding's Inequality.

**Theorem 5.4.** *Let $Z_1, \cdots, Z_T$ be independent random variables such that $Z_i \in [0, 1]$ for all $i \in [T]$. Let $\bar{Z} = \frac{1}{T}(Z_1 + \cdots + Z_T)$. We have $P(|\bar{Z} - E[\bar{Z}]| \geq \delta) \leq 2e^{-2T\delta^2}$ where $\delta \geq 0$.*

---

[4]If a job is sampled more than once, we can pretend that multiples copies of the same job are distinct and simulate round robin [25].

By Theorem 5.4 we have,

$$\Pr[X_1 < \epsilon | S_r|] = \Pr[X_1/|S_r| - a_1/n_r < \epsilon - a_1/n_r = \epsilon^2] \le 2\exp(-2|S_r|\epsilon^2) = 2/n^2.$$

Similarly, we can show that

$$\Pr[X_2 > \epsilon | S_r|] = 2/n^2$$

Thus, we conclude that $\Pr[B_1] \le 4/n^2$.

We consider the second bad event $|\tilde{y}_r - y_r| > \epsilon^2 n_r$, which we call $B_2$. Assume that $\tilde{q}_r$ is the size of a fixed job of rank in $[a_1, a_2]$. If $m'$ is the actual number of jobs of size $\tilde{q}_r$ in $J_{>r}$, $m_\ell = Binomial(m'/n_r, |S_r|)$. As before, using Theorem 5.4 we can show that $\Pr[B_2] \le 4/n^2$. Thus, we can avoid both bad events simultaneously with probability $1 - 8/n^2$.

Since we process each job in $S_r$ up to at most $\tilde{q}_r$ units, the total time we spend for estimation in Step 1 is at most $\tilde{q}_r \frac{1}{\epsilon^2} \log n$. This completes the proof.

### 5.1.2  Subprocedure: Step 2

We now elaborate on the second step. Our goal is to prove Lemma 5.3. To decide whether each sequence is good or not, as before we take a uniform sample from $S'_r$ of size $\frac{1}{\epsilon^2}(\log n + \log k)$ with replacement. By processing the jobs each up to $\tilde{q}_r$ units, we can decide the number of jobs in $S'_r$ of size no bigger than $\tilde{q}_r$. If the number is $c$, we estimate $\tilde{y}_r = \frac{c}{|S'_r|} n_r$. The proof follows the same lines as Lemma 5.2 and is omitted. The only minor difference is that we need to test if each of the $k$ sequences is good or not, and to avoid the bad events for all $k$ sequences simultaneously, we ensure the probability that the bad events occur for a sequence is at most $O(\frac{1}{kn^2})$. This is why we use a bigger sample size than in Step 1.

### 5.2  Analysis of the Algorithm's Performance

Let $\sigma^*$ be an arbitrary sequence against which we want to be competitive. The analysis proceeds in rounds. Let $b_r$ be the start time of round $r$; by definition $b_1 = 0$. For the sake of analysis we decompose our algorithm's cost as follows:

$$\sum_{r \in [L]} \left( \sum_{j \in J_r} (C_j - b_r) + T_r \cdot |J_{>r}| \right) + 2\mathrm{OPT}(J_{L+1}),$$

where $T_r$ is the total time spent in round $r$. To see this, observe that $\sum_{j \in J_r} (C_j - b_r)$ is the total completion time of jobs that complete in round $r$, ignoring their waiting time before $b_r$. Each job $j \in J_r$'s waiting time in the previous rounds is exactly $\sum_{r' \in [L-1]} T_{r'}$. The total waiting time over all of the jobs during rounds where they are not completed is $\sum_{r' \in [L-1]} T_{r'} \cdot |J_{>r'}|$. The last term $2\mathrm{OPT}(J_{L+1})$ follows from the fact that we use Round-Robin to finish the last few jobs in the final round $L + 1$.

To upper bound $A_r := \left( \sum_{j \in J_r} (C_j - b_r) + T_r \cdot |J_{>r}| \right)$ by $\mathrm{COST}(\sigma^*)$ we also decompose $\mathrm{COST}(\sigma^*)$ as

$$\sum_{r \in [L]} \left( \mathrm{COST}(\sigma^{*r+1}) - \mathrm{COST}(\sigma^{*r}) \right) + \mathrm{COST}(\sigma^{*L+1})$$

Recall that we complete jobs $J_r$ in round $r$ and $\sigma$ evolves from $\sigma^r$ to $\sigma^{r+1}$ by dropping $J_r$ from the sequence $\sigma^r$. Thus, it decreases the cost of following $\sigma$.

Since we use round-robin to finish the remaining jobs $J_{L+1}$ in the final round and round-robin is 2-competitive, our algorithm incurs cost at most $2\mathrm{OPT}(J_{L+1})$. If the assumption in Theorem 5.1 holds, this is at most $2\epsilon\mathrm{OPT}$.

Let's take a close look at $\left( \mathrm{COST}(\sigma^{*r+1}) - \mathrm{COST}(\sigma^{*r}) \right)$. This is equivalent to the following: For each pair $i \in J_r$ and $j \in J_{>r}$, $\sigma^*$ has an error $|p_i - p_j|$ if and only if it creates an inversion. This quantity can be charged generously. Let's call this aggregate error $\eta_r$. But $\mathrm{OPT}(J_r) + \sum_{i \in J_r, j \in J_{>r}} \min\{p_i, p_j\}$ should be charged sparingly. Let's call this part of optimum quantity $\mathrm{OPT}_r$. Note that we are using the following:

**Lemma 5.5.** $\text{OPT} = \sum_{r \in [L]} \left( \text{OPT}(J_r) + \sum_{i \in J_r, j \in J_{>r}} \min\{p_i, p_j\} \right) + \text{OPT}(J^{L+1})$

*Proof.* We know that the optimal schedule is Shortest-Job-First. Thus we have the following. Here we assume jobs are indexed in an arbitrary but fixed manner.

$$
\begin{aligned}
\text{OPT} &= \sum_{j \in J} p_j + \sum_{i \in J, p_i < p_j} p_i \\
&= \sum_{i \in J, j \in J, i \geq j} \min\{p_i, p_j\} \\
&= \sum_{r \in [L+1]} \left( \sum_{i \in J_r, j \in J_r \, i \geq j} \min\{p_i, p_j\} + \sum_{i \in J_r, j \in J_{>r}} \min\{p_i, p_j\} \right) \\
&= \sum_{r \in [L+1]} \left( \text{OPT}(J_r) + \sum_{i \in J_r, j \in J_{>r}} \min\{p_i, p_j\} \right) \\
&= \sum_{r \in [L]} \left( \text{OPT}(J_r) + \sum_{i \in J_r, j \in J_{>r}} \min\{p_i, p_j\} \right) + \text{OPT}(J^{L+1}) \quad \square
\end{aligned}
$$

Using the previous lemma, if we bound $A_r$ by $(1 + O(\epsilon))\text{OPT}_r + O(\frac{1}{\epsilon^5})\eta_r$ we will have Theorem 5.1 by scaling $\epsilon$ appropriately. Proving this for all $r \in [L]$ is the remaining goal.

We consider two cases.

**All Sequences Are Bad.** In this case we complete all jobs of size at most $\tilde{q}_r$. Further $\sigma^*_{r,\epsilon}$ has at least $\epsilon^2 n_r$ big jobs. This implies that there are at least $\epsilon^2 n_r$ big jobs that do not appear in $\sigma^*_{r,\epsilon}$. So, for each pair of such a big job and a non-big job, $\sigma^*_r$ creates an inversion and has an error of at least $\epsilon \tilde{q}_r$, which contributes to $\eta_r$. Thus, $\eta_r \geq \epsilon \tilde{q}_r \epsilon^4 n_r^2$.

Now we want to upper bound $A_r$. Since the delay due to estimation is at most $\tilde{q}_r \frac{2}{\epsilon^2}(\log n + \log k)$ and all jobs are processed up to $\tilde{q}_r$ units, we have $A_r \leq (\tilde{q}_r \frac{2}{\epsilon^2}(\log n + \log k)) * n_r + \tilde{q}_r(n_r)^2 \leq (2\epsilon + 1)\tilde{q}_r(n_r)^2$. Thus, $A_r \leq O(1)\frac{1}{\epsilon^5}\eta_r$.

**Some Sequences Are Good.** We will bound the expected cost of the algorithm for round $r$ when there is a good sequence. The bad event $B_1$ occurs with very small probability as shown in the analysis of Step 1. The contribution to the expected cost is negligible if the bad event occurs. Due to this, we may assume that the event $\neg B_1$ occurs.

Say the algorithm processes the first $\tilde{y}_r$ jobs in a good sequence $\sigma_r$. By Lemmas 5.2 and 5.3, $J_r$ processes all small and middle jobs in $\sigma^r_\epsilon$. Additionally, the algorithm may process up to $3\epsilon^2 n_r$ big jobs without completing them.

The total time it takes to process the big jobs is $4\epsilon^2 n_r \cdot \tilde{q}_r$ and up to $n_r$ jobs wait on them. The contribution to the objective of all jobs waiting while these are processed is at most $4\epsilon^2 n_r^2 \cdot \tilde{q}_r$. We call this the wasteful delay due to processing big jobs. We show that this delay is only $O(\epsilon)$ fraction of $A_r + A_{r+1}$.

Consider $A_r + A_{r+1}$. By definition of the algorithm, at least $\frac{1}{2}\epsilon n_r$ jobs of size at least $\tilde{q}_r$ are completed during rounds $r$ and $r+1$. If less than $\frac{1}{2}\epsilon n_r$ middle or big jobs complete in round $r$, we can show that $n_{r+1} \geq (1 - 2\epsilon)n_r$. Then, we observe that $J_{\geq r+1}$ must have at most $4\epsilon^2 n_r$ small jobs. This is because $\sigma^r_\epsilon$ includes at most $3\epsilon^2 n_r$ big jobs and it includes $\tilde{y}_r$ jobs. Since $y_r$ is the number of small and middle jobs and $|y_r - \tilde{y}_r| \leq \epsilon^2 n_r$, $\sigma^r_\epsilon$ must include all non-big jobs, except up to $4\epsilon^2 n_r$. Thus, most jobs completing in round $r+1$ are middle or big and we can show that the number of such jobs completing in round $r+1$ is at least $\frac{1}{2}\epsilon n_r$. Therefore, at least $\frac{1}{2}n_r$ jobs will wait on the first $\frac{1}{2}\epsilon n_r$ jobs of size at least $\tilde{q}_r$ completed. This implies, $A_r + A_{r+1} \geq \frac{1}{4}\tilde{q}_r \epsilon n_r^2$. This is at least a $\Theta(\frac{1}{\epsilon})$ factor larger than the wasteful delay.

Note that the delay due to processing big jobs and as well as the time spent computing the estimates (used in Lemma 5.2 and 5.3) is at most $n_r \cdot 6\epsilon^2 n_r \tilde{q}_r \leq O(\epsilon) \cdot (A_r + A_{r+1})$. In the following, we will bound the cost without incorporating these costs. Factoring in these two costs will increase the bound by at most $1/(1 - O(\epsilon))$ factor.

Thus, we only need to consider small and middle jobs that complete. For the sake of analysis, we can consider the worst case scenario where we first process middle jobs and then small jobs. We will bound $A_r$ by $\mathrm{OPT}(J_r)$ (i.e. without charging to $\eta_r$). In this case, $A_r/\mathrm{OPT}_r$ is maximized when when all small jobs have size 0. We will assume this in the following. Let $m$ be the number of mid sized jobs we complete. For brevity, assume that the number of small jobs is at most $\epsilon n_r$ although the actual bound is $\epsilon(1 + \epsilon)n_r$. Further, we assume that $m(m + 1) \approx m^2$ as we are willing to lose $(1 + \epsilon)$ factor in our bound. Let $n = n_r$ for notational convenience. Then, we have $A_r = m^2 \tilde{q}_r/2 + m\tilde{q}_r \epsilon n + (n(1 - \epsilon) - m)m\tilde{q}_r = m^2 \tilde{q}_r/2 + (n - m)m\tilde{q}_r$. In contrast, $\mathrm{OPT}_r = m^2 \tilde{q}_r/2 + ((1 - \epsilon)n - m)m\tilde{q}_r$. The ratio of the two quantities is $\frac{n - m/2}{(1-\epsilon)n - m/2}$ where $m \leq n$. Therefore, in the worst case $\frac{1/2}{1/2 - \epsilon} \leq (1 + 3\epsilon)$. Thus, $A_r$ can be bounded by $(1 + 3\epsilon)\mathrm{OPT}_r$.

## 5.3 Improved Guarantees

One can show that at least $(\epsilon - 4\epsilon^2)$ fraction of jobs complete in every round assuming no bad events occur: if all sequences are bad then all non-big jobs complete, which means at least $(\epsilon - \epsilon^2)n_r$ jobs complete due to Lemma 5.2. Otherwise, any good sequence $\sigma_r^\ell$ has at least $(\epsilon - \epsilon^2 - 3\epsilon^2)n_r$ non-big jobs in $\sigma_{r,\epsilon}^\ell$ due to Lemmas 5.2 and 5.3 and they all complete if $\sigma_r^\ell$ is chosen. Thus there are at most $O(\frac{1}{\epsilon} \log n)$ rounds and there are at most $O(\frac{k}{\epsilon} \log n)$ bad events, as described in Lemmas 5.2 and 5.3, to be avoided.

Suppose we run round robin all the time using $\epsilon$ of the speed, so even in the worst case we have a schedule that is $2/\epsilon$-competitive against the optimum and therefore against the best prediction as well. This will only increase our upper bound by $1/(1 - \epsilon)$ factor. Then, we can afford to have bad events with higher probabilities.

Specifically, we can reduce the sample size in Steps 1 and 2 to $s := \Theta(\frac{1}{\epsilon^2} \log(k(1/\epsilon^3) \log n))$ to avoid all bad events with probability at least $1 - \epsilon^2$; so if any bad events occur, at most an extra $(2/\epsilon) \cdot \epsilon^2 \mathrm{OPT}$ cost occurs in expectation. Then, we can show that we can do steps 1-4 as long as $n_r = \Omega(\frac{1}{\epsilon^4}(\log k + \log \log n + \log \frac{1}{\epsilon}))$.

Then the delay due to estimation is at most $2s\tilde{q}_r n_r$ in Steps 1 and 2. We want to ensure that this is at most $O(\epsilon)$ fraction of $A_r + A_{r+1}$. Recall that we showed $A_r + A_{r+1} \geq \frac{1}{4}\tilde{q}_r \epsilon n_r^2$. Thus, if we have $\frac{1}{4}\epsilon^2 \tilde{q}_r n_r^2 \geq 2s\tilde{q}_r n_r$, we will have the desired goal. This implies that all the delay due to estimation can be charged to $O(\epsilon)$ of the algorithm's objective as long as $n_r \geq 8\frac{1}{\epsilon^2}s$. This is why we switch to round-robin if $n_r \leq 8\frac{1}{\epsilon^2}s$.

## 5.4 Removing the Simplifying Assumption on Job Sizes

Recall that we chose $\rho$ uniformly at random from $[0, 1)$ and rounded up each $j$'s size to the closest number of the form $(1 + \epsilon)^{\rho+t}$ for some integer $t$. Although we then scaled down all job sizes by $(1 + \epsilon)^\rho$, we assume that we didn't do it. This assumption is wlog as all the bounds remain the same regardless of uniform scaling of job sizes.

Let $\eta^\ell$ be the prediction error of $\sigma^\ell$. Let $\bar{\eta}^\ell$ be the error after rounding up job sizes. Let $\bar{p}_j$ be $j$'s size after rounding. Note that i) $p_j \leq \bar{p}_j \leq (1 + \epsilon)p_j$; ii) if $p_i \leq p_j$, then $\bar{p}_i \leq \bar{p}_j$. The second property implies jobs relative ordering is preserved.

In the following we drop $k$ for notational convenience. We have

$$\eta := \sum_{i \neq j \in J} \mathbf{1}(p_i < p_j) \cdot \mathbf{1}(i \succ_{\sigma^*} j) \cdot |p_i - p_j|$$

$$\bar{\eta} := \sum_{i \neq j \in J} \mathbf{1}(p_i < p_j) \cdot \mathbf{1}(i \succ_{\sigma^*} j) \cdot |\bar{p}_i - \bar{p}_j|$$

**Lemma 5.6.** $\mathbb{E}\bar{\eta} = \Theta(1) \cdot \eta$.

*Proof.* Thanks to linearity of expectation it suffices to show that $\mathbb{E}|\bar{p}_i - \bar{p}_j| = \Theta(1) \cdot |p_i - p_j|$ for every pair of jobs $i$ and $j$.

Assume $p_i > p_j$ wlog. Scale both job sizes for convenience, so $p_j = 1$. Let $p_i/p_j = (1 + \epsilon)^\delta$.

Case 1. $\delta \geq 1$. In this case we have $\bar{p}_i > \bar{p}_j$ almost surely. Using the fact that rounding up increases job sizes by at most $1 + \epsilon$ factor, we can show that $(1/3)|p_i - p_j| \leq |\bar{p}_i - \bar{p}_j| \leq (1 + \epsilon)^2 |p_i - p_j|$.

Case 2. $\delta \in (1/2, 1]$. In this case we can show that $|p_i - p_j| = \Theta(\epsilon)$ and $\mathbb{E}|\bar{p}_i - \bar{p}_j| = \Theta(\epsilon)$.

Case 3. $\delta \in (0, 1/2)$. We have $|p_i - p_j| = (1 + \epsilon)^\delta - 1$ and $\mathbb{E}|\bar{p}_i - \bar{p}_j| = \int_{\rho=0}^{\delta} \epsilon(1 + \epsilon)^\rho d\rho = \frac{\epsilon}{\ln(1+\epsilon)}((1 + \epsilon)^\delta - 1)$. Thus, the ratio of the two is $\frac{\epsilon}{\ln(1+\epsilon)} = \Theta(1)$ for small $\epsilon > 0$. $\square$

What we showed was the following. For any $\bar{\eta}^\ell$, $\mathbb{E}A \leq (1 + \epsilon)(\text{OPT} + O(1)\frac{1}{\epsilon^5}\bar{\eta}^\ell) + 2(1 + \epsilon)\text{OPT}(J_{K+1})$. By taking expectation over randomized rounding, we have $\mathbb{E}A \leq (1 + \epsilon)^2(\text{OPT} + O(1)\frac{1}{\epsilon^5}\eta^\ell) + 2(1 + \epsilon)^2\text{OPT}(J_{K+1})$. By scaling $\epsilon$ appropriately, we obtain the same bound claimed in Theorem 5.1.

### 5.5 Learning $k$ Predicted Permutations

Now we show that learning the best $k$ permutations has polynomial sample complexity.

**Theorem 5.7.** *Let $\mathcal{D}$ be an unknown distribution of instances on $n$ jobs. Given $S$ independent samples from $\mathcal{D}$, there is an algorithm that outputs $k$ permutations $\hat{\sigma}_1, \hat{\sigma}_2, \ldots, \hat{\sigma}_k$ such that $\mathbb{E}_{J \sim \mathcal{D}}\left[\min_{\ell \in [k]} \eta(J, \hat{\sigma}_\ell)\right] \leq \min_{\sigma_1, \sigma_2, \ldots, \sigma_k} \mathbb{E}_{J \sim \mathcal{D}}\left[\min_{\ell \in [k]} \eta(J, \sigma_\ell)\right] + \epsilon$ with probability $1 - \delta$, where $S = \text{poly}(n, k, \frac{1}{\epsilon}, \frac{1}{\delta})$.*

*Proof.* The algorithm is basic ERM, and the polynomial sample complexity follows from Theorem 2.3 and Theorem 20 in Lindermayr and Megow [32]. $\square$

## 6 Conclusion

Despite the explosive recent work in algorithms with predictions, almost all of this work has assumed only a single prediction. In this paper we study algorithms with *multiple* machine-learned predictions, rather than just one. We study three different problems that have been well-studied in the single prediction setting but not with multiple predictions: faster algorithms for min-cost bipartite matching using learned duals, online load balancing with learned machine weights, and non-clairvoyant scheduling with order predictions. For all of the problems we design algorithms that can utilize multiple predictions, and show sample complexity bounds for learning the best set of $k$ predictions. Demonstrating the effectiveness of our algorithms (and the broader use of multiple predictions) empirically is an interesting direction for further work.

Surprisingly, we have shown that in some cases, using multiple predictions is essentially "free." For instance, in the case of min-cost perfect matching examining $k = O(\sqrt{n})$ predictions takes the same amount of time as one round of the Hungarian algorithm, but the number of rounds is determined by the quality of the *best* prediction. In contrast, for load balancing, using $k$ predictions always incurs an $O(\log k)$ cost, so using a constant number of predictions may be best. More generally, studying this trade-off between the cost and the benefit of multiple predictions for other problems remains an interesting and challenging open problem.

## Acknowledgments and Disclosure of Funding

Michael Dinitz was supported in part by NSF grant CCF-1909111. Sungjin Im was supported in part by NSF grants CCF-1617653, CCF-1844939 and CCF-2121745. Thomas Lavastida and Benjamin Moseley were supported in part by NSF grants CCF-1824303, CCF-1845146, CCF-2121744 and CMMI-1938909. Benjamin Moseley was additionally supported in part by a Google Research Award, an Infor Research Award, and a Carnegie Bosch Junior Faculty Chair.

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

# A Experiments

We now consider a preliminary empirical investigation of the algorithm proposed in Section 3 for min-cost perfect matching with a portfolio of predicted dual solutions. For this, we modify the experimental setup utilized by Dinitz et al. [18] to evaluate the case of single prediction for this problem. We use a training set which is a mixture between three distinct types of instances, and show that by using more than one prediction we see an overall improvement.

| Dataset | Shuttle | KDD | Skin [16] |
|---|---|---|---|
| # of Points ($n$) | 43500 | 98,942 | 100,000 |
| # of Features ($d$) | 10 | 38 | 4 |

Table 1: Datasets used in preliminary experiments.

**Experiment Setup and Datasets:** These experiments were performed on a machine with a 6 core, 3.7 GHz AMD Ryzen 5 5600x CPU and 16 GB of RAM. The algorithms were implemented in Python and the code is available at `https://github.com/tlavastida/PredictionPortfolios`.

To construct bipartite matching instances we adopt the same technique as Dinitz et al. [18] for Euclidean data sets. At a high level, their technique takes in a dataset $X$ of points in $\mathbb{R}^d$ and a parameter $n \in \mathbb{N}$, and outputs a distribution $\mathcal{D}(X, n)$ over dense instances of min-cost perfect matching with $n$ nodes on each side. Sampling from the distribution $\mathcal{D}(X, n)$ can be done efficiently. We consider three datasets (Shuttle, KDD, and Skin) from the UCI Machine Learning Repository [20] that were also considered in [18], see Table 1 for details.

Instead of considering each dataset (and its corresponding distribution) separately, we consider them together as a mixture model. For each dataset, we consider a sub-sample $X$ of 20,000 points and we set $n = 150$ (so $2n = 300$ nodes per instance) in order to construct each distribution $\mathcal{D}(X, n)$. We then sample 20 instances from each distribution and consider these 60 instances together as our training set (i.e. our learning algorithm doesn't know which dataset each instance was derived from). For testing, we sample an additional 10 instances from each dataset.

**Results:** To evaluate our approach, we vary the number of predicted dual solutions ($k$) learned from one (baseline) to five and also compare to the standard Hungarian method (referred to as "No Learning"). Following [18] our evaluation metrics are running time and the number of iterations of the Hungarian algorithm.

Given that the dataset has three distinct clusters by construction, we expect the average number of iterations of the Hungarian algorithm to decrease as $k$ grows from 1 to 3 and then stay stable. We also expect the running time to decrease as $k$ grows from 1 to 3, and then increase as $k$ grows from 3 to 5, as the cost of the projection step grows linearly with $k$.

In Table2 we have divided our results on the test instances by dataset, so that different scales don't obscure the results.

| Dataset | Shuttle | | KDD | | Skin [16] | |
|---|---|---|---|---|---|---|
| $k$ | # Iterations | Time (s) | # Iterations | Time (s) | # Iterations | Time (s) |
| No Learning | 149.0 | 0.929 | 304.4 | 1.97 | 63.1 | 0.396 |
| 1 | 77.9 | 0.601 | 149.4 | 1.09 | 68.7 | 0.472 |
| 2 | 59.0 | 0.433 | 144.0 | 1.08 | 306.7 | 2.690 |
| 3 | 42.4 | 0.350 | 144.0 | 1.09 | 38.9 | 0.318 |
| 4 | 42.4 | 0.373 | 130.3 | 1.05 | 38.9 | 0.342 |
| 5 | 42.4 | 0.394 | 136.1 | 1.15 | 38.9 | 0.357 |

Table 2: Average iteration count and average running time across each data set and value of $k$.

Observe that for the Shuffle dataset the process proceeds exactly as we had predicted – the number of iterations drops as $k$ grows from 0 to 3 and then stays constant, the wall clock time drops as well and then starts increasing.

For KDD and Shuffle the situation is more interesting. For KDD we don't see the nice inflection point at $k = 3$. We conjecture that the KDD dataset itself is diverse and induces many subclusters

for its family of matching instances, thus increasing the number of clusters keeps on improving the performance.

For the Skin dataset, we observe an anomaly at $k = 2$. We conjecture that this is due to the clustering in the learning stage allocating both predictions to the KDD and Shuffle datasets, essentially ignoring the Skin dataset and thus giving extremely poor performance. In other words, the gain from allocating the "extra" prediction to the other datasets was enough to outweigh the cost to the Skin data. This aligns well with our conjecture that the KDD data itself has many subclusters.

Overall, however, we see that the empirical evaluation supports our conclusion that there are performance gains to be had when judiciously using multiple advice models.