# OpenReview forum: "Algorithms with Prediction Portfolios"
_NeurIPS.cc/2022/Conference — NeurIPS 2022 Accept_

### Official Review · Reviewer_3fpq · 2022-07-03

**Rating:** 6
**Confidence:** 3
**Soundness:** 4 excellent
**Presentation:** 3 good
**Contribution:** 2 fair

**Summary:**

Over the last 10 years, a significant line of work has focused on how to augment optimization algorithms with predictions so that (a) improved performance is achieved when the predictions are accurate and (b) worst-case performance is recovered when the predictions are not accurate. In this work, the authors build on the more recent idea to augment optimization algorithms with $k$ predictions. For three canonical tasks, the authors quantify the computational cost (which they show is mostly quite small) to obtain equivalent performance to augmenting the optimization algorithm with the (unknown) best of the $k$ predictors.

**Questions:**

I've detailed this in the body of my review.

**Limitations:**

I see no negative societal impacts of this theoretical work.

**Strengths And Weaknesses:**

The paper is clearly written (although the supplementary material is hard to follow at first, due to interleaving theorem statements, proofs, and intuition like a textbook), and the problem well-motivated. The results are novel and the proofs all seem correct, although the technical contributions of the learning theory arguments are minimal. (For example, I believe Theorem 2.1 can be extracted from (the proofs of) more general results on compositions of VC classes. See Theorem 6.1 of https://arxiv.org/pdf/1105.4618.pdf, attributed to a 1997 textbook.)

The main limitation of the work is the incremental nature of the contribution and the utility of the results for demonstrating actual improved performance. For all three problems considered, the authors thoroughly cite how past work has considered the single predictor impact. The main contribution is improving (using the notation of Section 3, but the contribution is the same for all three problems)
$$
   \min_{\hat y} E_{(G,c)\sim\mathcal{D}} \ \lvert\lvert y^*(G,c) - \hat y \rvert\rvert_1
$$
to
$$
   \min_{\hat y^1,\dots,\hat y^k} E_{(G,c)\sim\mathcal{D}} \min_{\ell\in[k]} \  \lvert \lvert y^*(G,c) - \hat y^\ell \rvert\rvert_1.
$$
Obviously the second term is no larger than the first term. However, since there is a (sometimes small, but not zero) addition to computational complexity in order to achieve the second term, I think the main quantity of interest is *how much* smaller the second term gets. This should depend on the problem parameters, $k$, and some properties of $\mathcal{D}$ (e.g., how symmetric it is?). This will provide an answer to the question: is the improvement in performance worth the computational cost of considering $k$ predictors over a single predictor?

I do not mean to penalize the authors for their useful characterization of the existing literature, and hence I rate the paper as borderline. If the authors can quantify exactly how much better using $k$ predictors can be in terms of the relevant parameters for the three problems of interest, this would be sufficient to improve my rating to an accept.

---

> ### Author Response · Authors · 2022-08-02
> **Response to reviewer 3fpq**
>
> Q: I think the main quantity of interest is how much smaller the second term gets. This should depend on the problem parameters, , and some properties of  (e.g., how symmetric it is?). This will provide an answer to the question: is the improvement in performance worth the computational cost of considering  predictors over a single predictor?
>
> A:  This is a good point: we do not characterize theoretically when we get a significant improvement, and indeed this is a data dependent question. It is easy to construct  natural distributions and settings where we would expect to see significant improvement through using multiple predictions.  For example, consider the load balancing problem.  Say that instances of load balancing come from a clustered distribution where there is one cluster which represents high traffic days (e.g. weekdays) and one cluster for low traffic (e.g. weekends).  The learned weights from high traffic will be effectively useless for low traffic and vice versa.  Thus, having only one prediction, which will presumably be the “average” of the two clusters, will perform poorly for both of them. But if we have two predictions (one for low traffic and one for high) our results show that we can be as good as the best prediction, even though we do not know from which of the two clusters our input is coming from (extending the analogy, this deals with holidays that fall on weekdays).  This example shows that multiple predictions can lead to asymptotic improvements in algorithm performance.
>
> In other words, there are two different “types” of questions which are both interesting but are orthogonal: 1) Can we learn and utilize the best k predictions? And 2) Are the best k predictions sufficiently better than a single prediction to justify the increased computational cost?  This paper focuses on the first type of question, since the second type of question requires making distributional assumptions while the first type can be answered in a distributionally-agnostic way.  However, we point out that the increased computational cost to using multiple predictions is often small (see theorem statements and discussion to all reviewers), so even a modest improvement in performance would be worth it.

---

> > ### Comment · Reviewer_3fpq · 2022-08-07
> > **Score unchanged**
> >
> > Thanks to the authors for their comments. It was helpful of them to lay out clearly the cost dependence on k. They have engaged with my question, although I still think it may limit the utility of the results without such data-dependent guarantees of how much better it is to use k predictions, so I will keep my score the same.

---

> > > ### Author Response · Authors · 2022-08-08
> > > **Additional Example**
> > >
> > > Thank you for your response. Here is an (admittedly) simple data-dependent guarantee that can give more context. We note that this example can be extended to richer data models. This example holds for the matching error function and similar examples can be constructed for error functions used in load balancing and completion time scheduling.
> > >
> > > Suppose that the data comes from $k$ well separated gaussians: each with variance 1 and distance of $c$ between their means. When using a single prediction, the average error is going to be on the order of $c/2$ in the best case (when all $k$ gaussians are all equidistant from some central point). On the other hand, when using $k$ predictions, the average error is going to be O(1), due to the inherent variance of each gaussian.
> > >
> > > Now the benefit of using $k$ predictions depends precisely on the algorithm and the impact of $k$ on the overall cost. As we showed, for the matching problem, there is absolute benefit for any constant $c$ (as long as $k \leq \sqrt{n}$).  For load balancing (using a similar example) there is benefit as long as $c \geq \log k$, and so on.
> > >
> > > Thus we believe the key questions which our paper addresses, are  a) whether it is possible to be competitive with the *best* of the multiple predictions (so we don't have to know exactly which cluster an instance is from before we start working on it), and b) whether it is possible to learn the best portfolio of predictions.

---

> > > > ### Comment · Reviewer_3fpq · 2022-08-09
> > > > **Response to example**
> > > >
> > > > I am not convinced that this example answers my question. The authors claim that using $k$ predictions will lead to average error $O(1)$. However, this is only the case if each prediction is calibrated to a specific, separate cluster, which as far as I can tell is not guaranteed by any of the authors' results. Indeed, for the 3 examples they consider, it seems possible that all the predictions are just identical, and then there's no benefit. Note that the setting where you're given abstract predictors ahead of time (potentially each tuned to a specific cluster), and want to do as well as the best "pathwise expert" (i.e., the best one on each interaction) is captured already by the extensive literature on "phi"-regret; indeed, that's why the present work focuses on learning the predictions as well.

---

> > > > > ### Author Response · Authors · 2022-08-09
> > > > > **Additional response**
> > > > >
> > > > > Thank you for describing this additional concern. We point to the theory in Sections 3.3, 4.3, and 5.3 of the submitted paper and the supplementary: one of our key contributions is that we are able to learn nearly-optimal sets of predictions using clustering.  So if there *exists* a portfolio of k predictions which gives good performance, then we will find a portfolio that is nearly as good.  This implies that we will not learn something trivial like identical predictions if there is a better portfolio out there.
> > > > >
> > > > > To be slightly more precise for the given example of $k$ Gaussians in constant dimensions, the optimal solution is to place clusters in the middle of the $k$ Gaussians, which leads to constant expected cost. Therefore, using a constant-approximate algorithm for k-median (e.g. the result due to Byrka et al. with approximation ratio 2.675+$\epsilon$) will allow us to learn a portfolio of predictions with an average cost no more than a constant.  This is a significant improvement over the trivial single prediction or identical predictions.  So, in this particular example, it cannot be the case that the predictions will be identical (which is what the reviewer is concerned with) if the Gaussians are sufficiently separated.
> > > > >
> > > > > Reference:
> > > > > Jarosław Byrka, Thomas Pensyl, Bartosz Rybicki, Aravind Srinivasan, and Khoa Trinh. 2017. An Improved Approximation for k-Median and Positive Correlation in Budgeted Optimization. ACM Trans. Algorithms 13, 2, Article 23 (April 2017), 31 pages. https://doi.org/10.1145/2981561

---

> > > > > > ### Comment · Reviewer_3fpq · 2022-08-10
> > > > > > **Thanks for clarification**
> > > > > >
> > > > > > Ah, ok! This discussion and example has made clearer how to interpret the combination of results in each section (first one proving best prediction is recovered, second one proving oracle best k predictions is approximately recovered). I have updated my score from 5 to 6 in light of this, although I think the paper could benefit from an example or two like this spelled out.

---

### Official Review · Reviewer_Zbpu · 2022-07-04

**Rating:** 6
**Confidence:** 4
**Soundness:** 4 excellent
**Presentation:** 3 good
**Contribution:** 3 good

**Summary:**

The paper considers using multiple machine-learned predictions to improve classic algorithm design. In particular, the work is focused on three problem: min-cost bipartite matching, online makespan minimization, and non-clairvoyant scheduling. For the first problem, the goal is to improve run-time, whereas for the last two the goal is to improve competitive ratio.

In each setting, the paper gives an efficient algorithms that makes use of multiple predictions. When (at least one of) the predictions are good, the resulting learning-augmented algorithm can achieve better performance than classic, worst-case counterparts. Moreover, the paper works out the learnability conditions, showing that good predictions can be efficiently PAC learned from data.

**Questions:**

For matching, I am wondering if experimentally one can find multiple predictions can help (for real world or synthetic data). Let's say $k=5$ or $10$.  In principle it may not; that is, the best single prediction (in expectation over the distribution) is generally very good for every single instance in the distribution. My intuition is that if the distribution is somewhat "concentrated" such that the instances generally look alike, then multiple predictions may not help. On the other hand, if the distribution has a "multiple clusters" structure, then having multiple predictions can provide much better coverage and therefore improve performance a lot. Is it possible to formalize this intuitions and give some theory, for matching or graph problems in general?

The recent work by Chen, Silwal, Vakilian, and Zhang [Faster fundamental graph algorithms via learned predictions, ICML 22] shows that using (single) prediction can help to improve the run-time of graph algorithms, beyond the matching problem of Dinitz et al. I think the multiple prediction can be interesting more broadly in speeding up offline graph algorithms. This is another future direction the author(s) can explore.

For online makespan minimization, can the fractional solution be rounded in an online fashion? A majority of the paper by Lattanzi et al is spent on rounding (their Section 4 and 5).  I think their techniques can be applied here.

**Limitations:**

As I mentioned, the paper does not give a general framework of addressing multiple predictions in learning-based algorithms. In fact, reading its title, I expect they would.

Some experiments on matching could be very interesting.

**Strengths And Weaknesses:**

Strengths
-----
The paper is generally well-written, though with a few minor typos that may be confusing at times. I have listed them in a later section.

The results are novel in light of the recent literature on learning-based algorithms. In particular, this can be seen as a good follow-up on the question of algorithms with multiple predictions, recently initiated by Anand, Ge, Kumar, and Panigrahi ["Online algorithms with multiple
predictions" (ICML 22)]. As the author(s) point out,  Anand et al focus only on a set of online covering problems. The results of this paper do not lie within this framework. I believe this paper is a good addition to this line of work.

The theoretic claims are correct.

Weaknesses
----
In my view, some of the results in this work are a bit weak on a technical level. In particular, Section 3 on matching extends Dinitz et al in a fairly trivial fashion. The main algorithm (Algorithm 1) essentially says: take the largest dual prediction and run Dinitz et al single-predictor algorithm. This should be thought as a simple observation.  On the other hand, the algorithmic result of Section 4 on scheduling relies on several key insights of Lattanzi et al ("Online Scheduling via Learned Weights", SODA 20).  The learnability generally follows from a pseudodimension + PAC learning argument, again from Dinitz et al.

Compared with Anand et al, this paper does not provide a general framework. Rather, it addresses 3 separate problems of different nature. That is, matching is offline and the goal is to improve run-time, but the scheduling problems are online and the goal is to deal with the uncertainty.

Compared with Dinitz et al, the paper does not provide experiments.

Compared with Lattanzi et al, the paper does not address rounding fractional solutions.

Minor errors and missing references
-----
Line 11: “which prediction is [the] best”

Line 104: another paper that studies non-clairvoyant scheduling in single-predictor setting is “Optimal Robustness-Consistency Trade-offs for Learning-Augmented Online Algorithms” by Wei & Zhang (NeurIPS 20)

Line 249 reads a bit awkward, which is not making a definition. Maybe remove the “we say that”.

LHS of Line 10 of Algorithm 2 should be $S(j, \beta)$? I suggest the author(s) avoid overloading the notation of $A$ here.

LHS of Line 10 of Algorithm 2 should be $B$ instead of $A$.

Line 314: Optimal schedule, and Line 318: optimal objective value

Line 423 of the supplement: I suggest the author(s) to give a standard reference for constant-approximation of k-median; e.g., “A Constant-Factor Approximation Algorithm for the k-Median Problem” (https://www.sciencedirect.com/science/article/pii/S0022000002918829)

---

> ### Author Response · Authors · 2022-08-02
> **Response to reviewer Zbpu**
>
> Q: In my view, some of the results in this work are a bit weak on a technical level. In particular, Section 3 on matching extends Dinitz et al in a fairly trivial fashion.
>
> A: There are a variety of technical contributions in this paper.  The matching setting is particularly simple technically.  While Section 4 does indeed use many of the insights of Lattanzi et al., being able to compete with the *best* of the predictions does not immediately follow. The solution to learn multiple predictions and reduction to k-median require several new ideas not present in the literature. The results in Section 5 are technically complex, and we encourage the reviewer to consider all of the technical complexities present in the supplementary (full version).
>
> Q: For matching, I am wondering if experimentally one can find multiple predictions can help.
>
> A: Multiple predictions indeed can help.  See the discussion posted to all reviewers.
>
> Q: The recent work by Chen, Silwal, Vakilian, and Zhang [Faster fundamental graph algorithms via learned predictions, ICML 22] shows that using (single) prediction can help to improve the run-time of graph algorithms, beyond the matching problem of Dinitz et al. I think the multiple prediction can be interesting more broadly in speeding up offline graph algorithms. This is another future direction the author(s) can explore.
>
> A:  We completely agree with this comment.  The work by Chen et al. improves Dinitz et al. for the matching problem, gives similar results for the shortest paths problem, and shows how to reduce other problems to these problems.  We see no reason why the techniques in the first part of our paper, of learning and leveraging multiple predictions for matching, could not be extended to the results of Chen et al.   But we do not claim in this submission to have fully explored the space of multiple predictions.  Rather, we show that in three very different settings, it is possible to learn and use multiple predictions.  And our results on load balancing and scheduling are essentially orthogonal to the running-time based results of Dinitz et al. and Chen et al.
>
> Q: For online makespan minimization, can the fractional solution be rounded in an online fashion?
> A: Yes, it can.  The work of Lattanzi et al. and later Xian and Li (ICML 21) give algorithms to round the fractional solution online to an integral solution. This rounding can be done for any fractional solution, so, as mentioned in the paper (line 259), we will lose a small O(log log m) factor to convert to an integral solution.  This uses the best known rounding of Xian and Li.
>
> Q: The paper does not give a general framework of addressing multiple predictions in learning-based algorithms.
>
> A: As we see in the paper, using multiple predictions well requires algorithms and techniques tailored to the specific problem and setting.  After all, one would expect that the ideas that are useful for speeding up offline graph algorithms are very different than the ideas that are useful for improving competitive ratios of online algorithms!  Similarly, one would expect that different types of predictions require different techniques and ideas.  We have chosen these three problems because they are representative problems in the area of algorithms augmented with predictions, and we give new algorithmic techniques for each of these problems.  The bipartite matching problem is an offline algorithm and the metric of quality is *running time*. The load balancing and completion time problems are online and the metric of quality is the *competitive ratio*.  Load balancing and completion time further use different kinds of predictions; one is an ordering and the other is weights representing a proportional assignment.
>
> For these three different settings we develop techniques that we hope and believe can be used in other problems. For example, the ways that we learn and use multiple predictions for improving the run time of bipartite matching should be applicable to the recent work on improving the run time of graph algorithms by Chen, Silwal, Vakilian, and Zhang to obtain similar results for the shortest path problem and and other graph problems. Their algorithms’ run time is also parameterized by the error between actual optimum duals and learned duals like ours for weighted bipartite matching. Due to the structural similarities, we think it highly likely that our ideas could be paired with theirs to also give ways of using multiple predictions for other graph problems. This is an interesting direction for future research, but our goal in this paper was to show that multiple predictions can be learned and utilized in a variety of highly different settings.

---

### Official Review · Reviewer_vvHm · 2022-07-11

**Rating:** 6
**Confidence:** 3
**Soundness:** 3 good
**Presentation:** 3 good
**Contribution:** 3 good

**Summary:**

The paper attempts to examine solving certain online problems using a set of $k$ predictions and examine the statistical complexity of obtaining a good algorithm that uses such $k$ predictions. These problems include min-cost perfect matching, online load balancing and non-clairvoyant scheduling for total completion time. The task is to perform as well as the best prediction.


**Questions:**

Related Work: The related work is quite comprehensive. One of the related works that I found that the authors could contrast their work vis-a-vis learnability is :

Anand, Keerti, et al. "A regression approach to learning-augmented online algorithms." Advances in Neural Information Processing Systems 34 (2021): 30504-30517.

This is because in this work Anand et al did not consider the computational complexity of the optimization procedure that obtains the sample error minimizer. In comparison, the authors in this paper give a technique that is a polynomial time approximation algorithm for obtaining the predictions from the sample set that are O(1) within the best predictions obtainable from ERM.

Line 290: In this proof of Thm 3.4, won’t the algorithm need to look at all |S| choose k candidate solutions (and that would be exponential in $k$). The claim is that “we can find an O(1)-approximate solution which is of polynomial size in polynomial time”. What exactly in the time complexity and what is the procedure? Please clarify.


**Limitations:**

None.

**Strengths And Weaknesses:**

Originality: The paper’s work seems to be original. Using $k$ predictions to supplement online algorithms has been explored by multiple works in the past. In this paper, it has been extended to 3 other problems. The algorithmic techniques for Section 3 and Section 4 are fairly standard and known. Exploring the learnability of using such $k$ predictions seems to be a new undertaking. The algorithm for Section 5 also seems to be new work.

Clarity: The non-technical part of the paper is clearly written and easy to understand. Due to paucity of time, I could not verify the technical aspects of the paper but the theorem statements “make sense” to me. I found that the authors motivated the use of multiple predictions well.


Quality: The quality of results and writing is good. I did not check the math for the proofs in Section 5 but the algorithm seems intuitive. The paper is quite theoretical so having experiments/simulations is not warranted but would have been nice to see the proposed algorithm in Section 5 in action (even against synthetic datasets)

Significance: There is a plethora of work currently in Learning augmented algorithms. I find this work to be incremental in the field of using multiple predictions for Online Algorithms. The significance of the work is relevant since algorithms with predictions as a general field has gained so much traction in the community.

---

> ### Author Response · Authors · 2022-08-02
> **Response to reviewer vvHm**
>
> Q: One of the related works that I found that the authors could contrast their work vis-a-vis learnability is : Anand, Keerti, et al. "A regression approach to learning-augmented online algorithms." Advances in Neural Information Processing Systems 34 (2021): 30504-30517.
>
> A: This paper looks at using regression to learn predictions for online ski rental.  This is one of the first works to develop both an online algorithm that uses a prediction and shows how to learn the prediction.
>
> There are now several works that build on this idea to show how to learn parameters for learning augmented algorithms.  What is new in our paper is how to learn the $k$ best predictions.  Our learned predictions are proven to be near optimal. We develop new techniques to learn more than one prediction.  We further show how to leverage the best prediction for the given problem instance.
>
> Q: Won’t the algorithm need to look at all $|S|$ choose $k$ candidate solutions (and that would be exponential in $k$).
>
> A: While there are  $|S|$ choose $k$ candidate solutions, our algorithm does not need to iterate through all of them.  We are able to give an algorithm (leveraging k-median algorithms) to compute a near optimal solution in polynomial time. This procedure can be found in the supplementary material: Section 3.3 for Theorem 3.4, and Section 4.3 for Theorem 4.4 (which is Theorem 4.5 in the supplementary).

---

### Official Review · Reviewer_zg9d · 2022-07-11

**Rating:** 6
**Confidence:** 2
**Soundness:** 4 excellent
**Presentation:** 3 good
**Contribution:** 3 good

**Summary:**

Considering the fact that the worst-case performance of algorithms could be improved by equipping them with good machine-learned predictions, together with the increased use of "multiple" machine-learned models for solving a specific task, with each of them being specialized on a part of the problem, the paper tries to adapt algorithms for the case where multiple machine-learned predictors are used instead of a single one. They try to design algorithms for this case and bound the overhead caused compared to the single predictor case (since now the best predictor should be identified among the $k$ predictors) and the error that the algorithm incurs compared to the case where it is given the best predictor only. They investigate this problem for 3 fundamental problems being bipartite matching, load balancing and clairvoyant scheduling.

They start by proving that the complexity of looking for $k$ predictions is a factor of $k$ larger than that of a single predictor.

They further move to the three specific problems. First, they consider the min-cost bipartite matching problem with multiple predictions available. They show the running time of the state of the art algorithm for solving this problem for the case of single prediction is comparable to that of considering $k$ predictions when $k\leq O(\sqrt{n})$ which suggests that learning with $k$ predictions is coming for free in this case.

They further consider the problem of online load balancing (where a weight vector is to be learned to distribute a task between several machines with the goal of minimizing the maximum machine load) and show that there is a polynomial time algorithm that can output $k$ weight vectors which can minimize the expected minimum error with some bounded error which comes with a logaritmic cost with respect to $k$.

Finally they consider the problem of scheduling $n$ jobs on a machine when the time that each process takes for completion is not known prior to the completion of the process (clairvoyant scheduling). The goal is to minimize the sum of the completion times. They show that learning the best $k$ permutations is doable in polynomial time with bounded error compared to the best predictor.

**Questions:**

- How is the quality of the predictors affecting the performance of the final algorithm which takes into account all predictors? For each of the algorithms proposed, which of the following is more fabvorable? to have one very accurate and several very inaccurate predictors, or to have good performance in all predictors? Are the results also dependant on the performance of the worst predictor?

- Why are there no experiments conducted? It would be nice to see whether the running times in practice also match the ones stated in the paper or see the quality of the found solution compared to the algorithm using only the best predictor. Also having multiple machines with different performances and then analyzing the performance of the algorithm would be very interesting. Another set of interesting experiment is to vary $k$. I believe one can think of many more interesting experimental setups that could support the theory.

- Was there a reason for taking this 3 problems? How do you think one can generalize the result for other problems?

**Strengths And Weaknesses:**

Strengths:
- The paper considers a very well motivated problem.
- The paper is very carefully written.
- I believe there will be many applications and room for future work. Generalizing the results of the paper for any $k$ predictors solving any task and then bounding the error with respect to the error of each of the predictors would be very interesting.

Weaknesses:
- The supplementary does not match the main. Supplementary is an extended version of the paper and does not include only the proofs of the Theorems or Propositions stated in the main. Some of the mismatches: Theorem 2.1 in the main is Theorem 2.3 in the supplementary, Thorem 5.4 in the main is Theorem 5.7 in the supplementary etc.
- Section 2 is not self-contained. Without referring to the supplementary, it is almost impossible to understand what this section is trying to say or what the results are. Since most of the coming results in other sections are based on Theorem 2.1, I think it is important that enough details about this Theorem is provided in the main.
- The writing could be improved. Sometimes it is very hard to understand what you try to convey by the wording you use. I could not understand the sentence written lines 23-25. I also could not really understand what you try to do in the paper by only reading the abstract. More clear explanations would be great.
-Typo in line 25
- Although all the results are proven and there is no need for experiments to show that the theory works in practice, I believe experiments are missing from the paper and one could design many interesting experimental setups to show that the theory is working in practice.

---

> ### Author Response · Authors · 2022-08-02
> **Response to Reviewer zg9d**
>
> Q: The supplementary material has new numbering for the theorems.
>
> A: We will update the supplementary material to ensure consistency with the main body.  The supplementary was a full version of the paper and the numbering changed as we added new lemmas/theorems.  We apologize for the confusion.
>
> Q: How is the quality of the predictors affecting the performance of the final algorithm which takes into account all predictors? … Which of the following is more favorable: to have one very accurate and several very inaccurate predictors, or to have good performance in all predictors? Are the results also dependent on the performance of the worst predictor?
>
> A:  Each of the given $k$ predictions could be good for different types of instances (for example, if the problem instances come from a clustered distribution). Our main theoretical result is that our algorithms are able to leverage the *best* prediction for the given instance.  That is, we dynamically determine which prediction is best and use it.  The exact tradeoffs are given by the theorems in the paper, but our high-level results are that there is only a small cost to having more predictors, and we can perform as well as the best predictor.  Therefore, generally, it is better to have one highly accurate prediction and many inaccurate predictions than it is to have many many mediocre predictions.
>
> Q: Was there a reason for taking these 3 problems? How do you think one can generalize the result for other problems?
>
> A:  We have chosen these three problems because they are representative problems in the area of algorithms augmented with predictions. The paper demonstrates the utility of having more than one prediction in each of these different settings. The bipartite matching problem is an offline algorithm and the metric of quality is *running time*. The load balancing and completion time problems are online and the metric of quality is the *competitive ratio*.  Load balancing and completion time further use different kinds of predictions; one is an ordering and the other is weights representing a proportional assignment.
>
> For these three different settings we develop techniques that we believe can be used in other problems. For example, we believe that the predictions we use for improving the run time of bipartite matching can also be applied to the recent work on improving the run time of graph algorithms by Chen, Silwal, Vakilian, and Zhang to obtain similar benefits of multiple predictions for the shortest path problem and other graph problems.  We do not claim to have fully explored the possibilities of using a portfolio of predictions; rather, we have shown that on important and well-studied problems it is possible to both learn and leverage multiple predictions, and that the techniques are problem- and setting-dependent

---

> > ### Comment · Reviewer_zg9d · 2022-08-09
> > **Score Unchanged**
> >
> > I thank the authors for the detailed response and answers to some of my questions. My score remains unchanged since although some of my points are addressed, I believe the empirical results need to be added to the paper and peer reviewed before publication.

---

> > > ### Author Response · Authors · 2022-08-09
> > > **Response.**
> > >
> > > Thank you for reading our response. As we mentioned in the General Response, and illustrated in the simple model to reviewer 3fpq, our main contribution is developing algorithms that can take advantage of multiple predictions, namely to be competitive with the best prediction with minimal overhead in running time and solution quality. The empirical results we show in response to the reviewers serve to validate the theory and demonstrate that the algorithms are practical. However, we highlight that the goal of this paper is not to engineer improved practical performance, but rather to build the mathematical foundations for algorithms with multiple learned predictions.

---

### Author Response · Authors · 2022-08-02
**General Response**

We thank the reviewers for the helpful suggestions. Our submission studies three aspects of prediction portfolios: (i) how to use multiple predictions, (ii) how to learn the best set of multiple predictions, and (iii) the cost of using multiple predictions. We study these by focusing on a diverse set of representative problems.  These problems have different goals: running time (matchings), or competitive ratio (load balancing, scheduling).  They also have different types of predictions: duals, machine weights, and permutations. As we describe below, we believe our results can be easily extended to capture other problems in these areas.

Several reviewers asked about the cost of using multiple predictions. Part of our contribution is in proving theoretical bounds on this cost for each of the problems, this comes through as the dependence on $k$ in the analysis.

* For matchings (Theorem 3.3), we see a linear dependence on $k$ in the projection running time. That term is dominated by the cost of the Hungarian algorithm until $k$ reaches $\sqrt{n}$.
* For load balancing (Corollary 4.3) we see a logarithmic dependence on $k$ for the competitive ratio, hence there is a trade-off between more predictions and the quality of the best prediction.
* For scheduling (Theorem 5.1) we see a logarithmic dependence on $k$ in terms of the structure of the optimal solution, so here too there is a direct tradeoff.

While we believe that the analytical results are non-trivial and interesting in their own right, several reviewers asked for an empirical evaluation, to validate our findings.

**Empirical Evaluation**

While a full-blown empirical evaluation is out of scope for this comment, we can give at least a small validation.  We focus on the matching problem, previously studied by Dinitz et al. We will use a training set which is a mixture between three distinct types of instances, and show that by using more than one prediction we see an overall improvement.

To construct the dataset, we adapt the technique used by Dinitz et al. (See section 4 in their work). We take three datasets (KDD, Shuttle, Skin) and generate 20 bipartite matching instances on 300 nodes (150 nodes per side) from each dataset. Together, they constitute our training set. To test we generate an additional 10 instances of each kind.

To evaluate our approach, we vary the number of predictors learned from one (baseline) to five and also compare to the standard Hungarian method (no learning below). Following Dinitz et al. our evaluation metrics are running time and the number of iterations of the Hungarian algorithm.

Given that the dataset has three distinct clusters by construction, we expect the average number of iterations of the Hungarian algorithm to decrease as $k$ grows from 1 to 3 and then stay stable. We also expect the running time to decrease as $k$ grows from 1 to 3, and then increase as k grows from 3 to 5, as the cost of the projection step grows linearly with $k$.

We have divided our results on the test instances by dataset, so that different scales don’t obscure the results.

*Shuttle*
````{verbatim}
k			 |  avg number of iterations  | avg wall clock time |
No learning			149.0				0.929
1			    	77.9				0.601
2				59.0				0.433
3				42.4				0.350
4				42.4				0.373
5				42.4				0.394
````

*KDD*
````{verbatim}
k 			|  avg number of iterations  | avg wall clock time |
No learning			304.4			 	1.97
1				149.4				1.09
2				144.0				1.08
3				144.0				1.09
4				130.3				1.05
5				136.1				1.15
````

*Skin*
````{verbatim}
k			 |  avg number of iterations  | avg wall clock time |
No learning			63.1				0.396
1				68.7				0.472
2				306.7				2.690
3				38.9				0.318
4				38.9				0.342
5				38.9				0.357
````

**Analysis**

Observe that for the Shuffle dataset the process proceeds exactly as we had predicted – the number of iterations drops as k grows from 0 to 3 and then stays constant, the wall clock time drops as well and then starts increasing.

For KDD and Shuffle the situation is more interesting. For KDD we don’t see the nice inflection point at k=3. We conjecture that the KDD dataset itself is diverse and induces many subclusters for its family of matching instances, thus increasing the number of clusters keeps on improving the performance.

For the Skin dataset, we observe an anomaly at k=2. We conjecture that this is due to the clustering in the learning stage allocating both predictions to the KDD and Shuffle datasets, essentially ignoring the Skin dataset and thus giving extremely poor performance.  In other words, the gain from allocating the “extra” prediction to the other datasets was enough to outweigh the cost to the Skin data. This aligns well with our conjecture that the KDD data itself has many subclusters.

Overall, however, we see that the empirical evaluation supports our conclusion that there are performance gains to be had when judiciously using multiple advice models.

---

### Meta-Review · Area_Chair_iLRy · 2022-08-27

**Recommendation:** Accept
**Confidence:** Certain

**Metareview:**

The paper was generally well-received by the reviewers, who highlighted the significance of the setup, the strength of the results, and the enjoyable writing in their initial reviews. The additional experimental results provided during the rebuttal phase were particularly appreciated. Eventually we have reached consensus that the paper is clearly worthy of being published at NeurIPS 2022. I encourage the authors to work the additional experiments into the final version of the paper, and take all the remaining comments of the reviewers into account.

**Award:**

No

---

### Decision · Program_Chairs · 2022-09-14

Accept